# High-fidelity 3D live-cell nanoscopy through data-driven enhanced super-resolution radial fluctuation

Romain F. Laine[1,2,15,19], Hannah S. Heil [3,19], Simao Coelho [3], Jonathon Nixon-Abell [4,5], Angélique Jimenez[6], Theresa Wiesner[6], Damián Martínez[3], Tommaso Galgani[7,16], Louise Régnier [7], Aki Stubb[8,17], Gautier Follain[8,9], Samantha Webster[10], Jesse Goyette [10], Aurelien Dauphin [11], Audrey Salles [12], Siân Culley [1,18], Guillaume Jacquemet[8,9,13,14], Bassam Hajj [7] ✉, Christophe Leterrier [6] ✉ & Ricardo Henriques [1,2,3] ✉

Live-cell super-resolution microscopy enables the imaging of biological structure dynamics below the diffraction limit. Here we present enhanced super-resolution radial fluctuations (eSRRF), substantially improving image fidelity and resolution compared to the original SRRF method. eSRRF incorporates automated parameter optimization based on the data itself, giving insight into the trade-off between resolution and fidelity. We demonstrate eSRRF across a range of imaging modalities and biological systems. Notably, we extend eSRRF to three dimensions by combining it with multifocus microscopy. This realizes live-cell volumetric super-resolution imaging with an acquisition speed of ~1 volume per second. eSRRF provides an accessible super-resolution approach, maximizing information extraction across varied experimental conditions while minimizing artifacts. Its optimal parameter prediction strategy is generalizable, moving toward unbiased and optimized analyses in super-resolution microscopy.

Over the last two decades, super-resolution microscopy (SRM) developments have enabled the unprecedented observation of nanoscale structures in biological systems by light microscopy[1]. Stimulated emission depletion microscopy[2] has led to fast SRM on small fields of view with a resolution down to 40–50 nm. In contrast, structured illumination microscopy (SIM)[3] provides a doubling in resolution compared to wide-field (WF) imaging (~120 nm) with relatively high speed and large fields of view. Both super-resolution methods rely

[1]Laboratory for Molecular Cell Biology, University College London, London, UK. [2]The Francis Crick Institute, London, UK. [3]Optical Cell Biology, Instituto Gulbenkian de Ciência, Oeiras, Portugal. [4]Janelia Research Campus, Howard Hughes Medical Institute, Ashburn, VA, USA. [5]Cambridge Institute for Medical Research, Cambridge Univeristy, Cambridge, UK. [6]Aix-Marseille Université, CNRS, INP UMR7051, NeuroCyto, Marseille, France. [7]Laboratoire Physico-Chimie Curie, Institut Curie, PSL Research University, Sorbonne Université, CNRS UMR168, Paris, France. [8]Turku Bioscience Centre, University of Turku and Åbo Akademi University, Turku, Finland. [9]Faculty of Science and Engineering, Cell Biology, Åbo Akademi University, Turku, Finland. [10]EMBL Australia Node in Single Molecule Science, School of Biomedical Sciences, University of New South Wales, Sydney, New South Wales, Australia. [11]Unite Genetique et Biologie du Développement U934, PICT-IBiSA, Institut Curie, INSERM, CNRS, PSL Research University, Paris, France. [12]Institut Pasteur, Université Paris Cité, Unit of Technology and Service Photonic BioImaging (UTechS PBI), C2RT, Paris, France. [13]Turku Bioimaging, University of Turku and Åbo Akademi University, Turku, Finland. [14]InFLAMES Research Flagship Center, Åbo Akademi University, Turku, Finland. [15]Present address: Micrographia Bio, Translation and Innovation Hub, London, UK. [16]Present address: Revvity Signals, Tres Cantos, Madrid, Spain. [17]Present address: Department of Cell and Tissue Dynamics, Max Planck Institute for Molecular Biomedicine, Munster, Germany. [18]Present address: Randall Centre for Cell and Molecular Biophysics, King's College London, Guy's Campus, London, UK. [19]These authors contributed equally: Romain F. Laine, Hannah S. Heil. ✉e-mail: bassam.hajj@curie.fr; christophe.leterrier@univ-amu.fr; rjhenriques@igc.gulbenkian.pt

on complex optical systems to create specific illumination patterns. Single-molecule localization microscopy (SMLM) methods such as (direct) stochastic optical reconstruction microscopy ((d)STORM)[4,5], photo-activated localization microscopy[6] or DNA point accumulation in nanoscale topology (DNA-PAINT)[7,8], take a different approach, exploiting the stochastic ON/OFF switching capabilities of certain fluorescence-labeling systems. By separating single emitters in time and sequentially localizing their fluorescence signal, a near-molecular resolution (~10–20 nm) can be achieved; however, this commonly requires long acquisition times that range from minutes to days. Image processing and reconstruction tools, including multi-emitter fitting localization algorithms[9,10], Haar wavelet kernel (HAWK) analysis[11] or deep-learning assisted tools[12–14] reduce acquisition times by allowing for higher emitter density conditions. Alternatively, fluctuation-based approaches such as super-resolution radial fluctuations (SRRF)[15], super-resolution optical fluctuation imaging (SOFI)[16], Bayesian analysis of blinking and bleaching[17], multiple signal classification algorithm[18] or super-resolution with auto-correlation two-step deconvolution[19] can extract super-resolution information from diffraction-limited data (Supplementary Table 1). These fluctuation-based approaches only require subtle frame-to-frame intensity variations, rather than the discrete blinking events needed in SMLM, and as such, do not require high-illumination power densities. Thus, so long as images are acquired with a sufficiently high sampling rate to capture spatial and temporal intensity variations, these methods are compatible with most research-grade fluorescence microscopes. This makes them ideally suited for long-term live-cell SRM imaging[20].

In particular, SRRF is a versatile approach that achieves live-cell SRM on a wide range of available microscopy platforms with commonly used fluorescent protein tags[21]. It is now a widely used high-density reconstruction algorithm, as highlighted by an important uptake by the scientific community[22–25]. Since its inception, several adaptations of SRRF have been proposed by the community, such as those based on a combination with other advanced imaging approaches[26–28] or on the introduction of additional data preprocessing steps[29] (Supplementary Table 2 provides a summary), highlighting the interest in and potential impact of the method on the imaging community. The positive reception of the original SRRF method can also be attributed to its user-friendly and accessible implementation as a plugin for the Fiji framework[30]. In tandem, Andor Technology has also adapted an SRRF version for their camera-based imaging systems, including spinning disk confocal (SDC), a technology they named SRRF-Stream; however, obtaining optimal reconstruction results with any fluctuation-based method, including SRRF, can be challenging as they can suffer from reconstruction artifacts and lack signal linearity[31]. In an effort to start exploring these, we previously developed an approach for the detection and quantification of image artifacts termed SQUIRREL[32]. This tool has rapidly become a gold standard in the quantification of super-resolution image quality[33] by providing robust measures of how well the reconstruction corresponds to an enhanced resolution view of the equivalent diffraction-limited image. This comparison in turn aids in the identification of reconstruction artifacts. Through its image fidelity metrics, SQUIRREL provides an important platform to assist in the creation of new algorithms, such as those implementing deep-learning-based methods[34,35].

Obtaining three-dimensional (3D) SRM in live-cell microscopy still remains a challenge for the field; current implementations of 3D super-resolution methods come at the expense of a limited axial range and long acquisition times, often requiring major technical expertise[36–40]. Live-cell 3D SRM also generally requires a considerably higher illumination dose than two-dimensional (2D) super-resolution images. A feature that severely compromises cell health and viability[41]. Simultaneous multi-axial imaging when combined with super-resolution techniques, stands as an attractive alternative due to its capacity for near-instant volumetric imaging without discarding fluorophore emission, as would be the case in pinhole-based optical sectioning. This combination has been demonstrated before using image splitters[42], but with limitations associated with spherical aberrations arising from the optical geometry. Multifocus microscopy (MFM) provides an alternative 3D image acquisition framework, allowing multiple planes to be captured simultaneously (for example in this work, nine focal planes), while maintaining diffraction-limited image quality in every single plane[43–46].

Here, we present a new implementation of the SRRF approach, termed eSRRF, and highlight its improved capabilities in terms of image fidelity, resolution and user-friendliness. In eSRRF, we redefine some of SRRF's original fundamental principles to achieve improved image quality in the reconstructions. Our new implementation integrates the SQUIRREL engine to provide an automated exploration of the parameter space, yielding optimal reconstruction settings. This optimization is directly driven by the data itself and outlines the trade-offs between resolution and fidelity to the user. By highlighting the optimal parameter range and acquisition configurations, eSRRF minimizes artifacts and nonlinearity. Therefore, eSRRF improves overall image fidelity with respect to the underlying structure. The enhanced performance is verified over a wide range of emitter densities and imaging modalities.

We have additionally implemented the capability to achieve full 3D resolution improvement, bypassing the original SRRF's 2D capabilities. To do so, we adapted the method to analyze the nearly simultaneous multiplane acquisition of MFM, enabling the high-speed volume observation of fluorophore fluctuations. Here eSRRF benefits from the analysis of temporally coherent axial planes, meaning there is no time lag between axial planes. The estimation of radial fluctuations in 3D MFM data are based on the same reconstruction principles of 2D eSRRF, additionally assuming the z-axis point-spread-function (PSF) elongation. We also demonstrate a full implementation of 3D eSRRF with a comprehensive Fiji plugin and how it can facilitate fast 3D super-resolution imaging in living cells.

## Results

### eSRRF provides high-fidelity SRM images

Fluctuation-based SRM methods all suffer from the presence of artifacts and/or nonlinearity[31]. Here, we designed eSRRF with an emphasis on limiting reconstruction artifacts and maximizing image quality in the reconstruction of super-resolution images. The increased image fidelity results from the implementation of several new and optimized routines in the radial fluctuation analysis algorithm, introduced through a full rewriting of the code. In eSRRF, a raw image time series with fluctuating fluorescence signals is analyzed (Fig. 1a). First, each single frame is upsampled by interpolation (Fig. 1b). Here, in contrast to standard SRRF, we introduced a new interpolation strategy, exploiting a full data interpolation step based on Fourier transform before the gradient calculation. This approach outperforms the cubic spline interpolation employed in the original SRRF analysis by minimizing macro-pixel artifacts (Extended Data Fig. 1). Second, following the Fourier transform interpolation, intensity gradients $Gx$ and $Gy$ are calculated, and the corresponding weighting factor $W$ based on the user-defined radius $R$ is generated for each pixel. Based on gradient and weighting maps and the user-defined sensitivity parameter $S$ (Supplementary Table 3), the radial gradient convergence (RGC) is estimated. Thus, in the case of eSRRF, this estimation is not just based on a set number of points at a specific radial distance as it was handled by the previous implementation of SRRF, but over the relevant area around the emitter. This area and how each point contributes to the RGC metric is defined by the $W$ map. This allows us to cover the size of the PSF of the imaging system and thus, to exploit the local environment of the pixel of interest much more efficiently. Auto- or cross-correlation of the resulting RGC time series allows reconstructing a super-resolved image that shows high fidelity with respect to the underlying structure (Fig. 1b). Compared to the original SRRF, our new eSRRF approach demonstrates a clear

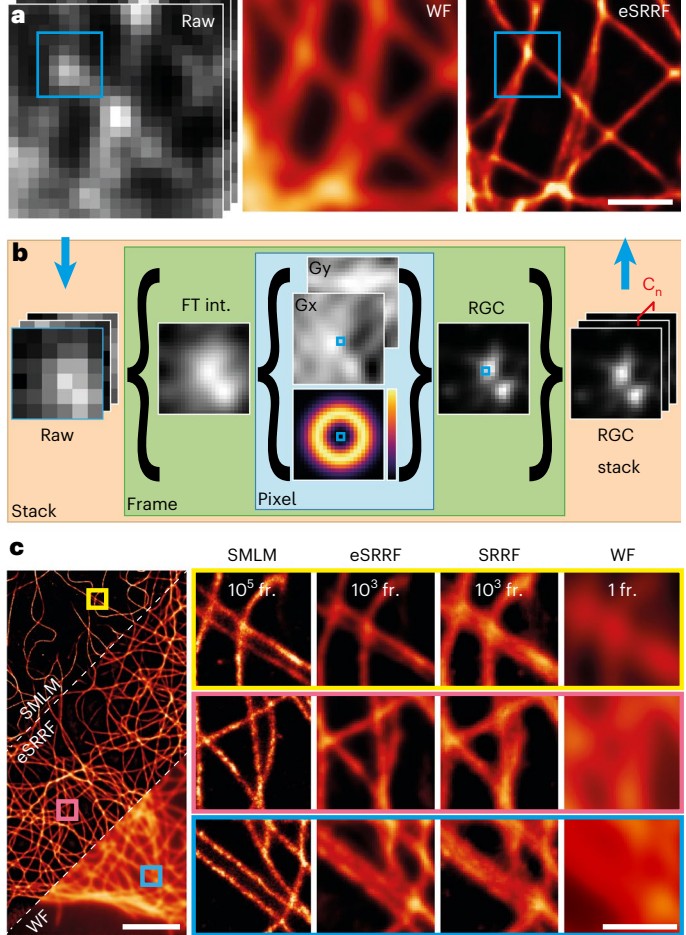

**Fig. 1 | eSRRF image reconstruction produces high-fidelity images. a**, eSRRF processing based on a raw data image stack (raw, left) of a microtubule network allows to surpass the diffraction-limited WF (middle) image resolution and to super-resolve features that were hidden before (eSRRF, right). **b**, eSRRF reconstruction steps. Each frame in the stack is interpolated (Fourier transform (FT) interpolation (int.)), from which the gradients $Gx$ and $Gy$ are calculated. The corresponding weighting factor map $W$ is created based on the set radius, $R$. Based on this, the RGC is calculated for each pixel to compute the RGC map. The RGC stack is then compressed into a super-resolution image by cross-correlation ($C_n$). **c**, Super-resolved reconstruction images from eSRRF and SRRF obtained from 1,000 frames of high-density fluctuation data (12.1 localizations per frame and $\mu m^2$), created in silico from an experimental sparse-emitter dataset (DNA-PAINT microscopy of immunolabeled microtubules in fixed COS-7 cells, 0.121 localizations per frame and $\mu m^2$). The SMLM reconstruction obtained from the sparse data and the WF equivalent are shown for comparison. The number of frames used for reconstruction is indicated in each column (FRC resolution estimate, SMLM 71 ± 2 nm, eSRRF 84 ± 11 nm, SRRF 112 ± 40 nm, WF 215 ± 20 nm). Scale bars, 1 μm (**a**, and insets in **c**) and 5 μm (**c**, left). FRC is shown as mean ± s.d.

improvement in image quality (Fig. 1c). Although these new implementations make eSRRF processing computationally more demanding, the implementation of OpenCL to parallelize calculations and minimize processing time allows the use of all available computing resources regardless of the platform[47].

To evaluate the fidelity and resolution of eSRRF with respect to the underlying structure, we performed analysis of a DNA-PAINT dataset with sparse localizations. For DNA-PAINT, standard SMLM localization algorithms applied to the raw, sparse data can provide an accurate representation of the underlying structure. By temporally binning the raw data, we generated a high-density dataset comparable to a typical live-cell imaging acquisition. Figure 1c shows the comparison of the

ground-truth (SMLM), eSRRF, SRRF and equivalent WF data. eSRRF is in good structural agreement with the ground truth and shows a clear resolution improvement over both the WF and the SRRF reconstruction. Line profiles reveal that eSRRF resolves features that were only visible in the SMLM reconstruction (Supplementary Fig. 1). We also estimated image resolution by Fourier ring correlation (FRC)[48] and decorrelation analysis[49]. Both provide a quantitative assessment of the resolution improvement of the different image reconstruction modalities (Supplementary Table 4). To facilitate a direct resolution comparison between SRRF and eSRRF, we obtained and analyzed images from a commercially available calibration standard, the Argo-SIM slide[50,51]. This slide contains structures of well-defined dimensions, purposely designed to test the resolution performance attainable by an optical system. By comparing eSRRF and SRRF analysis of Argo-SIM data, a higher resolving capacity of the eSRRF approach compared to its counterpart is demonstrated (Extended Data Fig. 2). Furthermore, the enhanced image fidelity recovered from eSRRF is quantitatively confirmed using SQUIRREL analysis on both simulated and experimental data (Supplementary Fig. 2, Extended Data Fig. 3 and Supplementary Note 1).

eSRRF not only achieves higher fidelity in image reconstruction than SRRF, but also provides a robust reconstruction method over a wide range of emitter densities. To estimate the range of emitter densities compatible with eSRRF, we again use low-density DNA-PAINT acquisitions and temporally binned the images with varying numbers of frames per bin. By increasing the number of frames per bin, the density of molecules in each binned frame increases. While the total number of molecules remains consistent throughout, this approach allows us to monitor the performance of eSRRF as a function of emitter density. Extended Data Fig. 4 presents the results from this analysis across the three temporal analyses provided (AVG, TAC2 and VAR; Supplementary Note 2 contains details on these temporal analyses). For each density range, a specific set of the processing parameters will provide the best image quality (Supplementary Table 3 and Supplementary Note 2) allowing access to high-fidelity super-resolved image reconstructions across a wide range of experimental conditions. At high emitter densities, eSRRF also outperforms the high-density emitter localization algorithm of ThunderSTORM[9] even in combination with HAWK analysis[11] (Supplementary Fig. 3). At the other extreme of sparse blinking densities typical of SMLM acquisitions, single-emitter fitting still provides unsurpassed localization precision and image resolution; however, it requires the processing of a large number of images. In this particular case of sparse blink dataset (typical SMLM data), eSRRF can provide a fast preview of the reconstructed image (Supplementary Fig. 4).

## eSRRF's reconstruction parameter exploration scheme

The decision to use a specific set of parameters for an image reconstruction is often based on user bias and expertise. This can lead to the inclusion of artifacts in the data[32]. To alleviate user bias and artifacts, here, we developed a quantitative reconstruction parameter search based on the concepts introduced by SQUIRREL. For this we compute visual maps of the FRC resolution and image fidelity as a function of radius $R$ and sensitivity $S$, exploring the eSRRF reconstruction parameter space. We use FRC to determine image resolution and a resolution-scaled Pearson (RSP) correlation coefficient as a metric for structural discrepancies between the reference and super-resolution images[32], here referred to as image fidelity. These two metrics do not necessarily correlate. The parameter sweep allows to explore trade-offs between FRC resolution and RSP fidelity (Supplementary Video 1) as a consequence of reconstruction parameter choice. To balance the two metrics, we use an F1 calculation to compute our quality and resolution (QnR) score:

$$QnR = \frac{2 \times RSP \times nFRC}{RSP + nFRC}$$

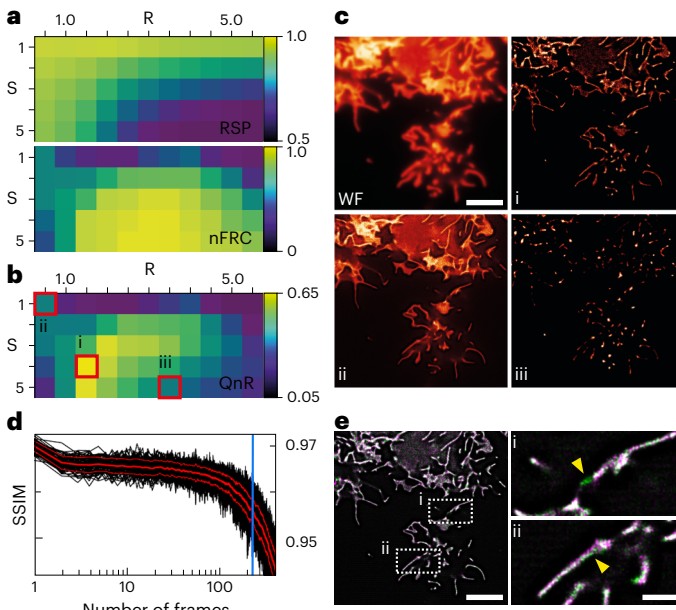

**Fig. 2 | eSRRF provides an automated reconstruction parameter search.**
**a**–**c**, Finding the optimal parameters to calculate the RGC. RSP and FRC resolution maps as functions of $R$ and $S$ reconstruction parameters for a live-cell TIRF imaging dataset published by Moeyaert et al.[53] (**a**). COS-7 cells are expressing the membrane targeting domain of Lyn kinase–SkylanS and were imaged at 33 Hz. Combined QnR metric map showing the compromise between fidelity and FRC resolution (**b**). WF image, optimal eSRRF reconstruction (i), $R = 1.5$, $S = 4$), low-resolution reconstruction (ii), $R = 0.5$, $S = 1$) and low-fidelity reconstruction (iii), $R = 3.5$, $S = 5$) (**c**). **d**,**e**, Estimating the optimal time window for the eSRRF temporal analysis based on tSSIM. The SSIM metric is observed over time, after ~200 frames it displays a sharp drop (**d**). The optimal time window is marked by the blue line. A color overlay of two consecutive reconstructed eSRRF frames with the optimal parameters and a frame window of 200 frames displays notable differences between the structures (marked by i and ii), which would lead to motion blurring in case of a longer frame window (**e**). Scale bars, 20 μm (**c**,**e**) and 5 μm (**e**–**i**(ii)).

Here *nFRC* is the normalized FRC resolution metric, ranging between 0 and 1, with 0 representing a poor resolution and 1 representing a high resolution. The QnR score ranges between 0 and 1, where scores close to 1 represent a good combination of FRC resolution and RSP fidelity, whereas a QnR score close to 0 represents a low-quality image reconstruction.

Figure 2 shows a representative dataset acquired with COS-7 cells expressing Lyn kinase–SkylanS, previously published by Moeyaerd et al.[52,53]. The eSRRF parameter scan analysis (Fig. 2a) shows how RSP fidelity and FRC resolution are affected by reconstruction parameters. RSP fidelity is high when using low sensitivity and/or low radius. In contrast, FRC resolution improves upon increasing the sensitivity over a large range of radii. This can be explained by the appearance of nonlinear artifacts at high sensitivity leading to low RSP fidelity but high FRC resolution. In addition, as the radius increases, the resolution of the reconstructed image decreases. The QnR metric map, shown in Fig. 2b, demonstrates that a balance can be found that leads to both a good resolution and a good fidelity. Figure 2c shows a range of image reconstruction parameters: the optimal reconstruction parameter set ($R = 1.5$, $S = 4$; Fig. 2c,i) and two other suboptimal parameter sets (Fig. 2c(ii),(iii)). Figure 2c(ii) shows a low-resolution image, whereas Fig. 2c(iii) has a high level of nonlinearity, the result of an inappropriately high sensitivity. While the QnR map can directly highlight optimal reconstruction settings by indicating the maximum QnR parameter combination, it also provides a window into the effect of reconstruction parameters on the output images to the user for critical analysis.

Local variations in background level, emitter density, and sample structure across the field of view can cause different reconstruction requirements and non-linearities in the QnR maps (Extended Data Fig. 5). User evaluation of QnR maps is an important component of this optimization strategy, and the parameter optimization tool is intentionally not designed to act as a black box. To aid in this evaluation, eSRRF lets users browse through the reconstructions associated with each QnR map value, enabling researchers to access the link between quality metrics and the corresponding variations in the reconstruction results. This makes eSRRF not only user-friendly but also ensures reproducible results with minimized user bias. In theory, this method could be used to improve the performance of any other image reconstruction algorithm. We have, for example, also tested applying the proposed parameter optimization to SRRF. Here, we observed that even with optimized parameters, SRRF reconstructions were not able to exceed the eSRRF performance (Extended Data Fig. 6).

An important aspect of live-cell super-resolution imaging is its capacity for observation and quantification of dynamic processes at the molecular level. With respect to fluctuation-based methods, there are two aspects to be considered when it comes to observing fast dynamics. As each super-resolved reconstruction is based on processing a stack of hundreds or more images, any dynamic changes happening within this frame window will lead to motion blur. On the other hand, reducing the number of frames for the super-resolved reconstruction can compromise the reconstruction quality. To address this aspect and provide an estimate of the optimal frame window, we have integrated temporal structural similarity (tSSIM) analysis into the eSRRF framework. Here, we calculate the progression of the structural similarity[54] at the different time points of the image stack relative to the first frame (Supplementary Fig. 5). This allows us to identify the local molecular dynamics (Supplementary Fig. 6) and estimate the maximum number of frames within which the structural similarity is retained, meaning that there is no observable movement (Fig. 2d). By combining tSSIM with eSRRF, we can determine an optimal number of frames required for the reconstruction to recover dynamics, while reducing motion-blur artifacts (Fig. 2e). The tSSIM is complemented by the parameter optimization tool, which jointly aids in finding the optimal radius and sensitivity parameter sets, and allows testing of different frame window sizes. This enables the user to identify the minimum number of frames to be analyzed or even acquired to ensure a good quality reconstruction.

## eSRRF works across a wide range of live-cell modalities

Here, we test our approach on a wide range of imaging modalities, including total internal reflection fluorescence (TIRF), fast highly inclined and laminated optical sheet (HiLO)-TIRF[55], SDC[56] and lattice light sheet (LLS) microscopy[57]. We show that eSRRF provides high-quality SRM images in living cells (Fig. 3). First, we imaged cultured neurons transiently expressing Skylan-NS tagged tubulin. Skylan-NS is an element of the photoswitchable fluorophore family, displaying a high level of fluorescence fluctuations that are ideal for eSRRF processing[58]. This allowed us to super-resolve the microtubule network (Fig. 3a). Traditional SRRF processing cannot tell apart microtubule bundles that are close to each other, but eSRRF can (Fig. 3a), even though they are tightly packed along the dendrites (Extended Data Fig. 5a). eSRRF can also be applied to volumetric datasets as obtained for example with LLS microscopy (Supplementary Video 2). Here, the eSRRF reconstruction of volumetric image stacks is obtained by processing each slice sequentially. Note that this approach can only effectively improve the lateral resolution (*xy* plane), while there is a sharpening comparable to deconvolution in the *z* direction, no resolution improvement over the diffraction-limited images should be expected. Figure 3b,c shows the plane by plane eSRRF processing of a LLS dataset of the ER in Jurkat cells allowing us to distinguish sub-diffraction-limited features along the *x* direction (Fig. 3b(i)),

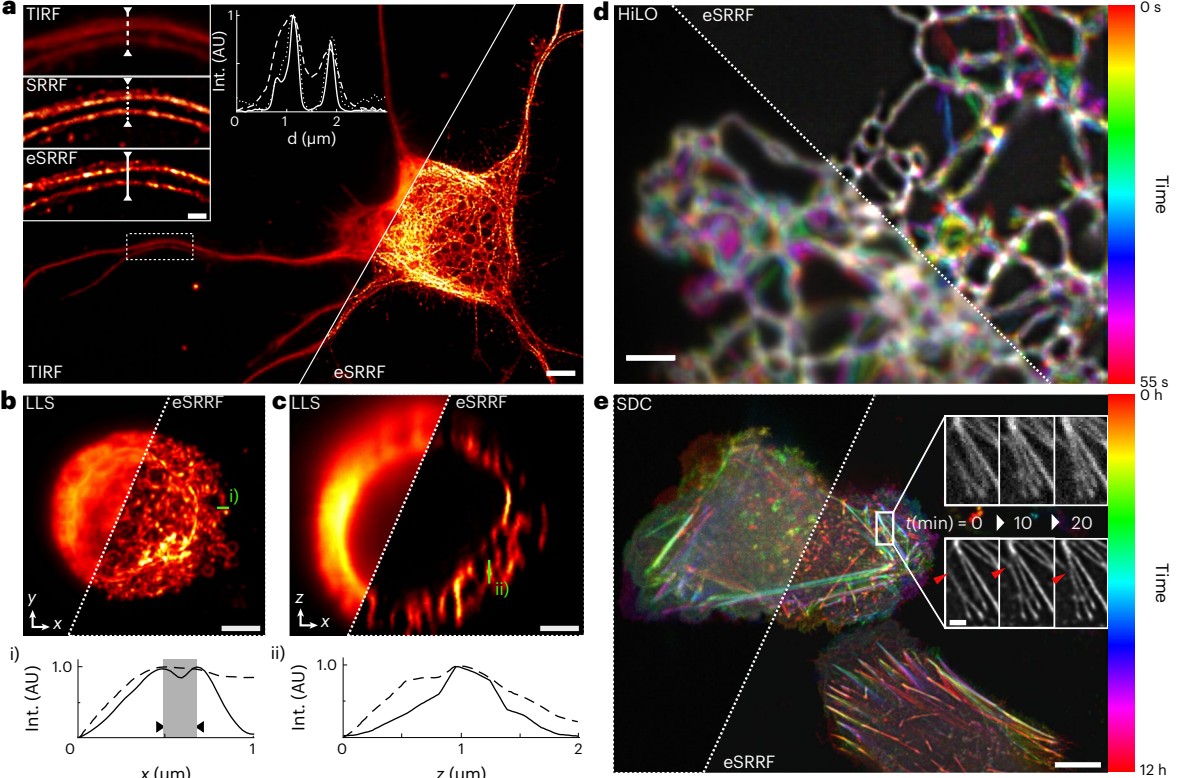

**Fig. 3 | Applications of eSRRF to a range of imaging modalities. a**, TIRF imaging of the microtubule network in a cultured neuron expressing Skylan-NS-tagged tubulin and subsequent eSRRF processing (see insets and line profile; TIRF-FRC, 425 ± 42 nm; SRRF-FRC, 213 ± 41 nm; eSRRF-FRC, 193 ± 51 nm). **b**,**c**, LLS imaging of the ER in Jurkat cells. *xy* projection (**b**) and *xz* projection using LLS microscopy (**c**) (top left) and the combination with eSRRF reconstruction (bottom right). As the acquisition is sequential, the eSRRF processing was applied on a slice-by-slice basis. Line profiles corresponding to the *x* and *z* direction are shown in i and ii, respectively. Sub-diffraction features separated by 190 nm in the lateral plane are marked in gray (FRC resolution LLS/eSRRF, 164 ± 9/84 ± 43 nm). AU, arbitrary units. **d**, Live-cell HiLO-TIRF of COS-7 cells expressing PrSS-mEmerald-KDEL

marking the ER lumen imaged at a temporal sampling rate of 10 Hz (temporal color-coding, FRC resolution HiLO/eSRRF, 254 ± 11/143 ± 56 nm). **e**, Live-cell SDC imaging of U2OS cells transiently expressing SkylanS−β-actin imaged over a time course of 12 h by acquiring substacks of 50 frames to generate a super-resolved eSRRF reconstruction at 10-min intervals (FRC resolution est. SDC/eSRRF, 484 ± 53 nm/151 ± 77 nm). Insets of three consecutive time points (top, SDC; bottom, eSRRF) show an example of actin bundles connecting and detaching (red arrow indicator) in the rectangular region marked by the white frame. Scale bars, 5 µm (**a**), 1 µm (**a** insets), 3 µm (**b**,**c**), 2 µm (**d**), 10 µm (**e**) and 2 µm (**e** insets). FRC shown as mean ± s.d.

but not in the *z* direction (Fig. 3c(ii)). While LLS can be seen as a fast and live-cell-friendly imaging approach, the sequential acquisition of a frame series at each axial plane generally slows down the acquisition to ~2 min per volume (79 × 55 × 35 µm³). Using HiLO-TIRF, the fast acquisition rates allowed us to track the ER network tagged with PrSS-mEmerald-KDEL in living COS-7 cells at super-resolution level (Fig. 3d). Here, a sampling rate of 10 Hz is achieved using rolling window analysis of eSRRF (Extended Data Fig. 7 and Supplementary Video 3). While TIRF and HiLO-TIRF imaging are set out for fast high-contrast imaging in close vicinity to the coverslip surface, SDC excels in fast and gentle in vivo imaging. Thus, with SDC imaging, we were able to record the dynamic rearrangement of SkylanS-tagged actin in U2OS cells over 12 h (Fig. 3e and Supplementary Video 4), showcasing the capacity of eSRRF to super-resolve living samples at low-intensity illumination. SDC also entails imaging far away from the coverslip and deep inside challenging samples as spheroids and live organisms, where eSRRF achieves enhanced performance as well (Extended Data Fig. 8).

## 3D live-cell super-resolution imaging by eSRRF and MFM
3D imaging capability is becoming increasingly important to understand molecular dynamics and interactions within the full context of their environment. In particular, obtaining true 3D SRM with improved resolution along the axial direction has recently become a key focus

of development in the field. Fluctuation-based SRM approaches have also been extended to 3D, notably SOFI[16,42] and more recently, random illumination microscopy[50], an approach that combines the concepts of fluctuation microscopy and the SIM demodulation principle; however, these and other 3D live-cell super-resolution approaches are still considerably hampered by their acquisition speed, a limitation that currently has only been surpassed by implementing deep-learning approaches[59] (Supplementary Table 5). To realize 3D eSRRF, we extended the algorithm to calculate the RGC in 3D (Supplementary Note 3). Consequently, we can reconstruct a volumetric image with enhanced resolution in the axial direction and in the lateral image plane. The approach was first validated with simulated 3D data (Extended Data Fig. 9). In these data, a woodpile structure can be seen, featuring filaments with axial distances ranging from 350 to 600 nm. These filaments are populated with single-molecule emitters, whose blinking events are set at increasing densities in each dataset. Their analysis allowed us to identify that, while at the highest densities only filaments crossing at an axial distance of 600 nm could be resolved, 3D eSRRF was able to distinguish filaments separated by under 400 nm in other density regimes. In practice, we implemented 3D eSRRF with an MFM system that is able to detect nine simultaneous axial planes on a single camera by using aberration-corrected diffractive optical elements[45] (Supplementary Fig. 7). By combining MFM and eSRRF, a

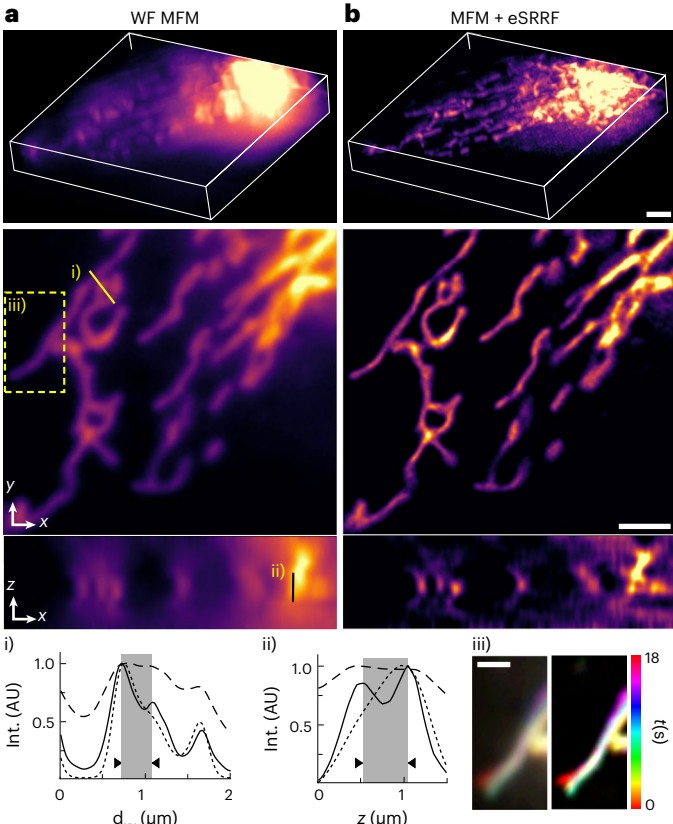

**Fig. 4 | eSRRF and MFM allows 3D live-cell super-resolution. a**, Live-cell volumetric imaging in MFM WF configuration of U2OS cells expressing TOM20-Halo, loaded with JF549. **b**, 3D eSRRF processing of the dataset creates a super-resolved volumetric view of $20 \times 20 \times 3.6\ \mu m^3$ at a rate of ~1 Hz (MFM + eSRRF). The 3D rendering (top); single cropped $z$-slice (FRC resolution in $xy$, interpolated, $231 \pm 10$ nm; eSRRF, $74 \pm 12$ nm) (middle); single cropped $y$-slice (FRC resolution in $xz$ eSRRF, $173 \pm 19$ nm) (bottom). (i) and (ii) mark the positions of the respective line profiles in the $xy$ and $z$-plane in the MFM (dashed line), deconvolved MFM (dotted line; Extended Data Fig. 10) and MFM + eSRRF (solid line) images (**a**,**b**). The distance of the structures resolved by eSRRF processing (marked gray) is 360 nm in the lateral directions ($x,y$) and 500 nm in the axial direction ($z$). (iii) marks the displayed area of the temporal color-coded projection of a single $z$-slice over the whole MFM (left) and MFM + eSRRF (right) acquisition. Scale bars, 2 μm (**a**,**b**) and 1 μm (iii). FRC shown as mean ± s.d.

super-resolved volumetric view ($20 \times 20 \times 3.6\ \mu m^3$) of the mitochondrial network architecture and dynamics in U2OS cells was acquired at a rate of ~1 Hz (Fig. 4). The eSRRF processing achieved super-resolution in lateral and axial dimensions, revealing sub-diffraction-limited structures. Figure 4 shows that eSRRF reveals structures that would otherwise remain undetected using conventional MFM (Fig. 4(i),(ii)). When compared to deconvolution analysis of the same data, the eSRRF reconstruction presents a higher resolution and the capacity to recover structural features of the sample that would otherwise remain hidden (Fig. 4(i),(ii) and Extended Data Fig. 10). Extending the 3D eSRRF processing to the full live-cell time lapse reveals rearrangement of the mitochondrial network at a super-resolution level (Fig. 4(iii)). To constrain cell damage, we maintained the observation time to ~20 s to minimize cellular damage. We were able to extend the observation time even more by using an extra dataset in which we reduced the excitation intensity by a factor of two. This second experiment achieves slightly lower resolution due to the reduced signal in these experimental conditions, while letting us observe mitochondrial network dynamics over the course of more than 3 min without observable signs of cell damage (Supplementary Video 5).

## Discussion

The new eSRRF approach builds on the previous capacity of SRRF, considerably improving image reconstruction quality and fidelity, as shown here for gold standards such as the Argo-SIM calibration structure[51], simulated data and the nuclear pore complex[60]. It showcases a new analytical engine for calculating the RGC transform, replacing the lower quality radiality transform of the original SRRF method. These modifications have also allowed us to extend the approach into full 3D super-resolution, by combining it with MFM, realizing fast 2D and 3D super-resolution imaging in live cells. While the performance of eSRRF may be surpassed in spatial or temporal resolution by deep-learning-based SRM approaches, these face a very specific set of limitations. Such deep-learning methods have been shown to convert sparse SMLM data[61] or even diffraction-limited images of dynamic structures[62] into super-resolved time series at rates above 50 Hz. Their extension to 3D isotropic volumetric live-cell SRM has demonstrated imaging rates of up to 17 Hz[59]; however, unlike eSRRF, these methods require previous knowledge of the dataset and generally require either simulated or experimental ground-truth data. These features also mean that insufficient, unbalanced or unsuitable data can lead to severe image degradation and hallucinations by deep-learning methods that are not easy to detect[63].

Here, we have introduced a data-driven parameter optimization approach that aids users in selecting optimal parameters learned directly from the data to be analyzed. These optimal parameters are chosen by balancing the need for high reconstruction fidelity together with high spatial and temporal resolution. In contrast to the training procedures employed in deep-learning approaches, the data-driven eSRRF parameter optimization bases its scoring on the direct unsupervised analysis of the data being collected via FRC and RSP calculations. As such, an eSRRF analysis is easy and reliable, especially when the data being collected have new properties or features that have not been seen before. Furthermore, the implemented parameter optimization based on the QnR metric directly provides dataset-specific insight into the relationship between image fidelity and resolution, enabling the user to critically analyze results and take an informed decision on analysis settings. By combining the QnR-based parameter sweep with a temporal window optimization, eSRRF achieves optimal spatial and temporal resolution while minimizing reconstruction artifacts and reducing user bias. This makes eSRRF a super-resolution method that shows users how to best analyze their data and gives them the information they need to find the best conditions for live-cell super-resolution imaging that is sensitive to phototoxicity. This provides new fundamental principles to make live-cell SRM more stable and reliable. While we developed and employed these original concepts in eSRRF, we expect this strategy to be easily transferable to other super-resolution methods that require an analytical component, as is the case for SMLM approaches.

To demonstrate the broad applicability of eSRRF, we showcase its application to a wide range of biological samples, from single cells to organisms, and imaging techniques from WF, TIRF, light sheet, SDC and SMLM imaging modalities. eSRRF shows robust performance over the different signal fluctuation dynamics displayed by various organic dyes and fluorescent proteins and over a wide range of marker densities, recovering high-fidelity super-resolution images even in challenging conditions in which single-molecule algorithms will fail. The enhanced performance of eSRRF will also benefit modifications of the original SRRF method that the community has previously proposed (Supplementary Table 2).

While eSRRF provides substantial improvements in image fidelity and resolution compared to the original SRRF implementation, there are still important caveats and limitations to consider. One limitation of eSRRF is for example the availability of bright, stable and live-cell-compatible fluorescent markers with emission characteristics that display suitable fluorescence fluctuations. Here, self-blinking organic dyes[64] and fluorogenic exchangeable HaloTag ligands[65] show

promise as high-performance probes for eSRRF. Furthermore, while 3D eSRRF still needs specialized hardware, we envision that future implementations may become more compatible with readily available commercial optical systems. In addition, it may be possible to explore gentler and faster MFM approaches[66]. The computational complexity of eSRRF makes processing times longer compared to SRRF. Additionally, finding the optimal balance between resolution and fidelity relies on user evaluation of the parameter space, so there is still some subjectivity in selecting final parameters. At low emitter densities, single-molecule localization methods can still provide better resolution than eSRRF. There are also challenges with observing very fast dynamics below the temporal resolution of the image stack used for reconstruction. A further machine-learning accelerated version of eSRRF is now being developed and available in the NanoPyx Python-based framework, which also includes a napari plugin[67]. Future advances could automate the parameter optimization process more fully using machine learning. Overall, while current implementations still have limitations, the development of eSRRF demonstrates the potential for data-driven optimization to improve image reconstruction quality and accessibility in SRM. eSRRF is currently implemented as an open-source GPU-accelerated Fiji plugin, accompanied by a detailed user guide (https://github.com/HenriquesLab/NanoJ-eSRRF) as well as in napari and Python, making it widely available to the bioimaging community.

## Online content

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

## Methods

The imaging conditions and eSRRF-processing parameters for each dataset are summarized in Supplementary Table 4.

### Fluorescence microscopy simulations

The field of simulated fluorescent molecule distribution was simulated over a 5-nm resolution grid with ImageJ 1.45 f and the NanoJ-eSRRF>Fluorescence simulator application. As a ground truth, fan pattern emitters were placed on concentric rings with radii increasing by 220 nm steps. On each ring the molecules were separated by 57.5, 115, 173, 230, 288 and 345 nm, respectively. Each emitter was allowed to blink independently with an on/off rate of $100 \, s^{-1}$ and $50 \, s^{-1}$, respectively over the entire acquisition without bleaching (500 frames at 10-ms exposure). Based on this distribution, the fluorescence image with 100 nm pixel size was created by convolution with a Gaussian kernel with $\sigma = 0.21 \lambda/NA$, as suggested by Zhang et al.[68], where $\lambda$ is the emission wavelength (here 580 nm) and NA is the numerical aperture of the microscope (here NA = 1.4). A realistic Poisson photon noise and a Gaussian read-out noise were added to the images to simulate an experimental dataset.

The diffusing particle datasets were generated as single emitters represented by a Gaussian PSF and with Gaussian noise moving at a constant speed, with a Python script available as a GoogleCoLabs Jupyter notebook on GitHub at https://github.com/HenriquesLab/NanoJ-eSRRF.

The MFM simulations of stacked lines were performed using a custom-written framework NanoJ-TheSims in ImageJ. Ground-truth images were provided as 3D stacks on an upsampled grid (5-nm pixel size in (*xy*) and 10-nm slice separation). For 3D multiplane simulations, a PSF stack was generated with the ImageJ plugin PSFGenerator[69] using the Born and Wolf model, 1.4 NA, and emission wavelength of 650 nm, with pixel sizes and plane separations matching the ground truth. For each molecule (nonzero pixel) in the ground-truth stack, the time for the molecule to undergo permanent photobleaching was randomly chosen from an exponential distribution with the rate parameter k_bleach. The time series describing transitions between on and off states was described as a two-state model, with the time spent in each state determined by random selection from exponential distributions with rates k_on (off to on) and k_off (on to off). On/off transitions were generated until the total length of the time trace reached the bleaching time (which was only permitted to occur from an on state). For the simulations used here, rate parameters were k_bleach = $0.077 \, s^{-1}$, k_on = $0.026 \, s^{-1}$ and k_off = $1.82 \, s^{-1}$. These were binned into discrete time traces per the camera settings of exposure time = 10 ms and read time = 5 ms, allowing for fractional appearances of molecules which undergo a transition within a frame under the assumption of an emission rate of 15 photons $s^{-1}$. For each *z* plane, an image stack of length *n* frames covering the simulated experiment time (typically 100 s) was created and populated with the number of emitted photons per molecule per frame from the discrete time traces. For 2D simulations, this stack was then convolved with a 2D Gaussian. For 3D multiplane simulations, for every molecule appearance, the slice of the PSF stack corresponding to the *z* position of the plane being simulated was multiplied by the number of emitted photons and then added onto the upsampled grid, centered on the molecule location. For all simulations, Poisson noise was added to the stack, the grid binned to 100 nm 'camera' pixels and Gaussian read noise was added. Based on the resulting nine image planes, an MFM image stack was reconstructed and a 3D image stack was reconstructed with the following eSRRF parameters: *M* = 4, *R* = 3 and *S* = 6, VAR. Deconvolution of the interpolated 3D image stack was performed with the ImageJ plugin DeconvolutionLab2 (ref. 70) with the PSF used for the simulation and the Richardson–Lucy algorithm (40 iterations). To increase the emitter density, the frames were binned and averaged.

### DNA-PAINT of microtubule network

COS-7 cells (ATCC CRL-1651) were cultured in phenol-free DMEM (Gibco) supplemented with 2 mM GlutaMAX (Gibco), $50 \, U \, ml^{-1}$ penicillin, $50 \, \mu g \, ml^{-1}$ streptomycin (Penstrep, Gibco) and 10% fetal bovine serum (FBS; Gibco) at 37 °C in a humidified incubator with 5% $CO_2$. Cells were seeded on ultraclean[71] 18-mm diameter thickness 1.5 H coverslips (Marienfeld) at a density of $0.3–0.9 \times 10^5$ cells per $cm^2$.

Cells were fixed and stained according to previously published protocols[72]. Cells were extracted at 37 °C for 45 s in 0.25% Triton-X (T8787, Sigma), 0.1% glutaraldehyde in the cytoskeleton-preserving buffer 'PIPES-EGTA-Magnesium' (PEM; 80 mM PIPES pH 6.8, 5 mM EGTA and 2 mM $MgCl_2$) followed by 10 min in 0.25% Triton-X and 0.5% glutaraldehyde in PEM. After a 7-min quenching step with a fresh solution of 0.1% $NaBH_4$ in phosphate buffer at room temperature, cells were permeabilized and blocked for 1.5 h at room temperature in blocking buffer (phosphate buffer 0.1 M, pH 7.3, 0.22% gelatin (G9391, Sigma) and 0.1% Triton-X-100).

Primary antibody labeling was performed at 4 °C overnight with a mix of two anti-α-tubulin mouse monoclonal IgG1 antibodies (DM1A (T6199, Sigma) and B-5-1-2 (T5168, Sigma)) diluted 1:300 in blocking buffer. After $3 \times 10$-min washes with blocking buffer, the cells were incubated with a goat anti-mouse antibody conjugated to a DNA sequence (P1 docking strand, Ultivue kit) for 1 h at room temperature and diluted 1:100 in blocking buffer. After incubation, cells were washed with blocking buffer for 10 min and $2 \times 10$ min with phosphate buffer.

DNA-PAINT imaging was performed on an N-STORM microscope (Nikon) equipped with 647-nm lasers (125 mW at the optical fiber output). After injection of an 0.25-nM imager strand (I1-ATTO655, Ultivue) in 500 mM NaCl in 0.1 M PBS, pH 7.2, buffer, 50,000 frames were acquired at 60% power of the 647-nm laser with an exposure time of 30 ms per frame to obtain low-density ground-truth data.

Image reconstruction was performed with the ImageJ/Fiji plugins NanoJ-SRRF, NanoJ-eSRRF, ThunderSTORM and HAWK. To create datasets with increased emitter density, temporal binning was performed by summing substacks of the SMLM image sequence. The raw data had an average emitter density of 0.121 localizations per frame and $\mu m^2$, which remained constant throughout the whole DNA-PAINT acquisition.

### SIM imaging of the Argo-SIM calibration sample

Super-resolution 3D-SIM imaging was performed on a Zeiss ELYRA PS.1 microscope (Carl Zeiss) using an αPlan-Apochromat ×100/1.46 oil immersion objective and an EMCCD Andor 887 1K camera. For SIM reconstruction, 15 images (five phases and three rotations) of the gradually spaced line pattern on the Argo-SIM calibration slide (Argolight) were acquired. SIM image reconstruction was performed with the software ZEN Black v.11.0.2.190 (Carl Zeiss).

### Live-cell imaging of cultured neurons

**Neuronal culture and transfection.** All procedures were in agreement with the guidelines established by the European Animal Care and Use Committee (86/609/CEE) and were approved by the Aix-Marseille University ethics committee (agreement no. G13O555). Primary neuronal cell culture was produced by extracting hippocampi from E18 rat pups of both sexes from pregnant female Wistar rats (Janvier Labs). Hippocampi were dissected and homogenized by trypsin treatment followed by mechanical trituration and seeded on 18-mm diameter, round, no. 1.5H coverslips at a density of 30,000 cells $cm^{-2}$ for 3 h in serum-containing plating medium (MEM with 10% FBS, 0.6% added glucose, $0.08 \, mg \, ml^{-1}$ sodium pyruvate and $100 \, UI \, ml^{-1}$ penstrep). Coverslips were then transferred, cells down, to Petri dishes containing confluent glia cultures conditioned in NB+ medium (neurobasal medium supplemented with 2% B27, $100 \, UI \, ml^{-1}$ penstrep and $2.5 \, \mu g \, ml^{-1}$ amphotericin) and cultured in these dishes (Banker method[73]).

Neurons were transfected with either 1 µg pEGFP–α-tubulin (Clontech cat. no. 632349) or pSkylan-NS–α-tubulin at 6–9 d in culture

with Lipofectamine (2 μl, Life Technologies). pSkylan-NS–α-tubulin was created by exchanging the mScarlet sequence in the mScarlet–α-tubulin (Addgene plasmid #85045[74]) to Skylan-NS (Addgene plasmid #86785[58]) by Gibson assembly[75] with a 0.005 U μl⁻¹ T5 exonuclease (M0663S, NEB), 0.033 U μl⁻¹ Phusion DNA polymerase (F-553L, Finnzymes) and 5.33 U μl⁻¹ TAQ DNA ligase (MB42601, NZYTEch) master mix. Live-cell imaging was performed 24–48 h later. Before live imaging, neurons were transferred to Hibernate medium E (Brainbits), supplemented with 2% B27, 2 mM Glutamax and 0.4% D-glucose, and maintained in a humid chamber at 36 °C for the duration of the experiments (Okolab).

**TIRF microscopy.** Live imaging of neurons expressing pSkylan-NS–α-tubulin were performed on an inverted microscope ECLIPSE Ti2-E (Nikon Instruments) equipped with an ORCA-Fusion sCMOS camera (Hamamatsu Photonics K.K., C14440-20UP) and a CFI SR HP Apochromat TIRF ×100 oil (NA 1.49) objective. Samples were sequentially illuminated with laser light at 405 nm and 488 nm for 100 ms at 5% laser power, respectively with an active Nikon Perfect Focus System and with the NIS-Elements AR 5.30.05 software (Nikon).

**2D-SIM microscopy.** Live-imaging experiments of neurons expressing pEGFP–α-tubulin were performed on a Nikon N-SIM S structured illumination microscope with a Nikon Ti inverted microscope using a CFI Apochromat TIRF ×100 C Oil (NA 1.49) objective and an ORCA-Fusion BT camera (Hamamatsu Photonics K.K., C15440-20UP). The sample was excited using laser light at 488 nm for 200 ms, at 70% laser power and kept at 37 °C using a Tokai Hit STX stage-top incubator with active Nikon Perfect Focus System. For each SIM image, nine raw images (three phases and three orientations) were acquired and reconstructed using NIS-Element 5.30.02 (Nikon).

### LLS sample preparation and acquisition

The ER of Jurkat cells (Cellbank Australia Jurkat-ILA1) was stained with BODIPY ER-Tracker (Thermo Fisher Scientifc, E34250) and incubated on a poly-L-lysine-coated (P4707, Sigma) 5-mm round coverslip at 37 °C. The cells were fixed for 15 min at 37 °C with 4% PFA (15710, Electron Microscopy Sciences) in a buffer with 10 mM MES (pH 6.1, M3671, Sigma), 5 mM EGTA (E4378, Sigma), 5 mM fresh glucose (49159, Sigma-Aldrich), 150 mM NaCl (AJA465, Ajax Finechem) and 3 mM MgCl₂ (pH 7.0, MA029, Chem-Supply) followed by two washing steps with the buffer. The LLS data were acquired using a commercially available 3i LLS[57] running SlideBook v.6 (Intelligent Imaging Innovations). LLSM has two orthogonal objective lenses, a 0.71 NA, 3.74-mm LWD water-immersion illumination objective and a 1.1 NA, 2-mm LWD water-immersion imaging objective, matched to the light sheet thickness for optimal optical sectioning creating an evenly illuminated plane of interest, which enables high spatiotemporal resolution (230 × 230 × 370 nm in *xyz*). We used a light sheet under a square lattice configuration in dithered mode. Images were acquired with a Hamamatsu ORCA-Flash 4.0 camera. Each plane of imaged volume was exposed for 10 ms with a 642-nm laser. The sample was imaged on a piezo stage with the dithered light sheet moving at 276 nm step size in the *z* axis. To create an eSRRF 'frame', a time lapse of 100 frames were taken per axial plane at 7.6 mHz per volume (79 × 55 × 35 μm³). After eSRRF processing, the images were deskewed with the LLSM Fiji plugin.

### Live-cell HiLO-TIRF microscopy

COS-7 cells (ATCC CRL-1651) were grown in phenol red-free DMEM supplemented with 10% (*v/v*) FBS (Gibco), 2 mM L-glutamine (Thermo), 100 U ml⁻¹ penicillin and 100 μg ml⁻¹ streptomycin (Thermo) at 37 °C and 5% CO₂. The 25-mm no. 1.5 coverslips (Warner Scientific) were pre-cleaned by (1) a 12-h sonication in 0.1% Hellmanex (Z805939, Sigma); (2) five washes in 300 ml distilled water; (3) a 12-h sonication in distilled water; (4) an additional round of five washes in distilled water; and (5) sterilization in 200-proof ethanol and were allowed to air dry.

Coverslips were coated with 500 μg ml⁻¹ phenol red-free Matrigel (356237, Corning). Cells were seeded at 60% confluency. Transfections were performed using Fugene6 (E2691, Promega) according to the manufacturer's instructions. Each coverslip was transfected with 750 ng of PrSS-mEmerald-KDEL to label the ER structure and with 250 ng of HaloTag-Sec61b-TA (not labeled with ligand for these experiments).

Imaging was performed using a customized inverted Nikon Ti-E microscope outfitted with a live-imaging chamber to maintain temperature, CO₂ and relative humidity during imaging (Tokai Hit). The sample was illuminated with a fiber-coupled 488-nm laser (Agilent Technologies) through a rear-mount TIRF illuminator. Imaging was performed such that the TIRF angle was manually adjusted below the critical angle to ensure HiLO illuminations and that the ER was captured within the illumination plane. The average power density over the full illumination field was 123 mW cm⁻². Fluorescence was collected using a ×100 α-Plan-Apochromat 1.49 NA oil objective (Nikon) using a 525/50 filter (Chroma) placed before an iXon3 EMCCD (DU-897; Andor). Imaging was performed with 5-ms exposure times for 60 s. The precise timing of each frame was monitored using an oscilloscope directly coupled into the system (mean frame rate ≈ 95 Hz).

### Spinning disk confocal sample preparation and acquisition

**SkylanS–β-actin U2OS cells.** U2OS osteosarcoma cells were grown in DMEM (Sigma, D1152) supplemented with 10% FBS (S1860, Biowest). U2OS cells were purchased from DSMZ (Leibniz Institute DSMZ-German Collection of Microorganisms and Cell Cultures, ACC 785). U2OS cells were transfected with 1 μg SkylanS-(GGGGS)x3–β-actin plasmid (Addgene plasmid #128938)[76] using Lipofectamine 3000 (L3000008, Thermo Fisher Scientific) according to the manufacturer's instructions. To image the actin, dynamic, transfected cells were plated on high-tolerance glass-bottom dishes (MatTek Corporation, coverslip no. 1.7) pre-coated first with poly-L-lysine (10 μg ml⁻¹ for 1 h at 37 °C) and then with bovine plasma fibronectin (10 μg ml⁻¹ for 2 h at 37 °C). The SDC microscope used was a Marianas spinning disk imaging system with a Yokogawa CSU-W1 scanning unit on an inverted Zeiss Axio Observer Z1 microscope controlled by SlideBook v.6 (Intelligent Imaging Innovations). Images were acquired using a Photometrics Evolve, back-illuminated EMCCD camera (512 × 512 pixels) and a ×100 (NA 1.4 oil, Plan-Apochromat) objective (Carl Zeiss). For long-term live-cell imaging, 50-fr substacks were acquired at 10-min intervals with a ABBAABB sequence switching between A, 1 ms of 405-nm laser activation and B, 25 ms of 488-nm laser excitation.

To culture cells on polyacrylamide gel, U2OS cells expressing endogenously tagged paxillin-GFP[22] were cultured as described in the previous section. Cells were left to spread on ~9.6 kPa polyacrylamide gel and were imaged using a SDC microscope with a ×63 objective (NA 1.15 water, LD C-Apochromat) objective (Zeiss) and an acquisition time of 100 ms. One hundred frames were used for the eSRRF reconstruction. The parameter sweep option as well as SQUIRREL analyses (resolution-scaled error and RSP values), integrated within eSRRF, were used to define the optimal reconstruction parameters.

The spheroids were based on MCF10DCIS.com (DCIS.com) lifeact-RFP cells[77] cultured in a 1:1 mix of DMEM (D1152, Sigma) and F12 (51651C, Sigma) supplemented with 5% horse serum (16050-122; GIBCO BRL), 20 ng ml⁻¹ human EGF (E9644; Sigma-Aldrich), 0.5 mg ml⁻¹ hydrocortisone (H0888-1G; Sigma-Aldrich), 100 ng ml⁻¹ cholera toxin (C8052-1MG; Sigma-Aldrich), 10 μg ml⁻¹ insulin (I9278-5ML; Sigma-Aldrich) and 1% (*v/v*) penstrep (P0781-100ML; Sigma-Aldrich). Parental DCIS.com cells were provided by J.F. Marshall (Barts Cancer Institute, Queen Mary University of London). To form spheroids, DCIS.com cells expressing lifeact-RFP were seeded as single cells, in standard growth medium, at low density (3,000 cells per well) on growth factor-reduced Matrigel-coated glass-bottom dishes (coverslip no. 0; MatTek). After 12 h, the medium was replaced by a normal growth medium supplemented with 2% (*v/v*) growth factor-reduced Matrigel and 10 μg ml⁻¹ of

FITC-collagen (type I collagen from bovine skin, C4361, Merck). After 3 d, spheroids were fixed with 4% PFA for 10 min at room temperature and imaged using a SDC microscope. The microscope used, as well as the image processing, are as described in the previous section.

Zebrafish (*Danio rerio*) housing and experimentation were performed under license MMM/465/712-93 (issued by the Ministry of Agriculture and Forestry, Finland). Transgenic zebrafish embryos expressing mcherryCAAX in the endothelium (genotype *Tg*(*KDR:mcherryCAAX*)) were cultured at 28.5 °C in E3 medium (5 mM NaCl, 0.17 mM KCl, 0.33 mM CaCl$_2$ and 0.33 mM MgSO$_4$). At 2 d after fertilization, embryos were mounted in 0.7% low-melting-point agarose on glass-bottom dishes. Agarose was overlaid with E3 medium supplemented with 160 mg l$^{-1}$ tricaine (E10521, Sigma). Imaging was performed at 28.5 °C using a SDC microscope. The microscope used, as well as the image processing, are as described in the previous section with the exception that 150 frames were used for the eSRRF reconstruction.

### Multifocus microscopy sample preparation and acquisition

HeLa cells (ATCC CRM-CCL-2) were cultured in complete medium (DMEM (11880, Thermo Fisher Scientific) + 1% Glutamax + 1% penstrep supplemented with 10% FBS (26140079, Thermo Fisher Scientific)) and transfected with TOM20 (translocase of outer mitochondrial membrane) fused to HaloTag. TOM20–HaloTag was labeled with Janelia fluor 549 HaloTag ligand (GA1110, Promega) by incubating the dye at 10 nM in DMEM medium for 15 min at 37 °C. MFM imaging was performed in DMEM without phenol red medium. The MFM setup used was described in detail by Hajj et al.[44], where excitation was performed with the 555 nm line of a Lumencor Spectra light engine and imaging was performed using a Nikon Plan Apo ×100 oil immersion objective with NA 1.4. Images of all nine focal planes were captured on an Andor DU-897 EMCCD camera at a rate of 20 ms per frame. The focus offset *dz* was 390 nm between consecutive focal planes.

The 3D image registration was performed based on multicolor fluorescent beads (TetraSpeck Fluorescent Microspheres kit; T14792, Invitrogen), immobilized on a coverslip. Images of the beads were recorded while axially displacing the sample with a z-step size of 60 nm.

To overlay and align the nine focal planes, a calibration table was created based on the bead images with the NanoJ-eSRRF plugin tool 'Get spatial registration from MFM data' (NanoJ-eSRRF>Tools>Get spatial registration from MFM data). This spatial registration was applied to the live-cell MFM data during the 3D eSRRF processing (a detailed manual can be found at https://github.com/HenriquesLab/NanoJ-eSRRF/wiki).

To extract the shape of the MFM PSF in the different focal planes (Supplementary Fig. 7), the 3D PSF was extracted from the bead reference with the respective NanoJ-eSRRF plugin tool (NanoJ-eSRRF>Tools>Extract 3D PSF from stack).

Deconvolution was performed with the classic maximum likelihood estimation algorithm in the Huygens Professional v.21.10 (Scientific Volume Imaging, The Netherlands, http://svi.nl). The 3D-rendered images were created with napari[78].

### Estimation of image resolution

To estimate the image resolution based on FRC with the NanoJ-SQIRREL ImageJ plugin[32], the raw time-series image stacks were split into even and odd frames. The independent image sequences were analyzed with SRRF, eSRRF or ThunderSTORM and based on the resulting pairs of processed images, the resolution was estimated by FRC reporting the mean and s.d. of the resolution in equally sized subregions of the image. Image resolution was also assessed by decorrelation analysis with the ImageJ plugin ImageDecorrelationAnalysis[49].

### Statistics and reproducibility

Figures show representative data from 4 (Fig. 1, Extended Data Fig. 2 and Supplementary Fig. 1), 55 (Fig. 2a–c), 2 (Figs. 2d,e and 3b–e and Extended Data Figs. 5a, 7 and 8a), 3 (Figs. 4 and 3a, Extended Data Figs. 3, 5b and 10 and Supplementary Figs. 2, 3, 4 and 6), 24 (Extended Data Figs. 1 and 8b), 8 (Extended Data Fig. 4), 90 (Extended Data Fig. 6), 6 (Extended Data Figs. 1 and 8c) and 7 (Extended Data Fig. 9) independent experiments.

### Reporting summary

Further information on research design is available in the Nature Portfolio Reporting Summary linked to this article.

### Data availability

The datasets are available on *Zenodo* at https://doi.org/10.5281/zenodo.6466472 (ref. 79).

### Code availability

eSRRF is available as a supplementary software or can be accessed from the GitHub page at https://github.com/HenriquesLab/NanoJ-eSRRF. This resource is fully open-source and includes a Wiki manual.

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

### Acknowledgements

We thank J. Lippincott-Schwartz and C. Obara at Janelia Farm for their assistance with the dynamic ER-TIRF data and for reading the paper. This work was supported by the Gulbenkian Foundation (H.S.H., S. Coelho, D.M. and R.H.) and received funding from the European Research Council under the European Union's Horizon 2020 research and innovation program (grant agreement no. 101001332 to R.H. and D.M.), the European Commission through the Horizon Europe program (AI4LIFE project with grant agreement 101057970-AI4LIFE

and RT-SuperES project with grant agreement 101099654-RT-SuperES to R.H.), the European Molecular Biology Organization (EMBO-2020-IG-4734 to R.H. and ALTF 499-2021 and Scientific Exchange Grant 9958 to H.S.H.), the Wellcome Trust (203276/Z/16/Z to R.H.), the Fundação para a Ciência e Tecnologia, Portugal (FCT fellowship CEECIND/01480/2021 to H.S.H., CEECIND/07466/2022 and PTDC/08248/2022 to S. Coelho), the Chan Zuckerberg Initiative Visual Proteomics Grant (vpi-0000000044 to R.H.), InnOValley Proof of Concept Fund (IOVPoC-2021-01 to S. Coelho) and National Health and Medical Research Council of Australia (no. APP1183588 to S. Coelho and J.G.). R.F.L. acknowledges the support of the MRC Skills Development Fellowship (MR/T027924/1). T.G. acknowledges the support by the European Union's Horizon 2020 research and Innovation program under the Marie Sklodowska-Curie Grant Agreement No. 666003 and the Fondation pour la Recherche Médicale under the Fin de these 2020 program (FDT202001010813). UTechS PBI is part of the France-BioImaging infrastructure network and A. Salles and A.D. are supported by the French National Research Agency (ANR-10-INBS-04; Investments for the Future) and acknowledge support from ANR/FBI and the Région Ile-de-France (program 'Domaine d'Intérêt Majeur-Malinf') for the use of the Zeiss LSM 780 Elyra PS1 microscope. S. Culley acknowledges support from a Royal Society University Research Fellowship (URF\R1\211329). This study was supported by the Academy of Finland (G.J., 338537 and G.F., 332402), the Sigrid Juselius Foundation (G.J.), the Cancer Society of Finland (G.J.), the Åbo Akademi University Research Foundation (G.J., CoE CellMech), the Drug Discovery and Diagnostics strategic funding to Åbo Akademi University (G.J.) and by the InFLAMES Flagship Program of the Academy of Finland (decision no. 337531). The Cell Imaging and Cytometry Core facility (Turku Bioscience, University of Turku, Åbo Akademi University and Biocenter Finland) and Turku Bioimaging are acknowledged for services, instrumentation and expertise. C.L. acknowledges funding from the Agence National de la Recherche (ANR-20-CE13-0024). C.L. acknowledges the INP NCIS imaging facility and Nikon Center of Excellence for Neuro-NanoImaging for service and expertise, with funding from Excellence Initiative of Aix-Marseille University, A*MIDEX, a French 'Investissements d'Avenir' program (AMX-19-IET-002) through the Marseille Imaging and NeuroMarseille Institute. B.H. acknowledges funding from the Fondation pour la Recherche Médicale (DEI20151234398), the Agence National de la Recherche (ANR-19-CE42-0003-01), the LabEx Cell(n)Scale (ANR-11-LABX-0038, ANR-10-IDEX-0001-02), the Institut Curie, Agence pour la Recherche sur le Cancer (ARC Foundation), DIM ELICIT and from ITMO Cancer of Aviesan on funds administered by Inserm (grant no. 20CP092-00). B.H. recognizes the support of France-BioImaging infrastructure grant ANR-10-INBS-04 (Investments for the Future).

## Author contributions

R.F.L., B.H., C.L and R.H. designed the study. R.F.L., S. Culley and R.H. wrote the algorithm. S. Culley wrote the script for data simulation. R.F.L., S. Coelho, H.S.H., J.N.A., A.J., T.G., A. Stubb, G.F., S.W, J.G., G.J., B.H., L.R., T.W., D.M., A. Salles, A.D. and C.L. performed the experiments. H.S.H. tested, documented and validated the algorithm and analyzed the data. S. Coelho analyzed the LLS data. All authors planned experiments and contributed to the writing of the paper.

## Competing interests

The authors declare no competing interests.

## Additional information

**Extended data** is available for this paper at https://doi.org/10.1038/s41592-023-02057-w.

**Correspondence and requests for materials** should be addressed to Bassam Hajj, Christophe Leterrier or Ricardo Henriques.

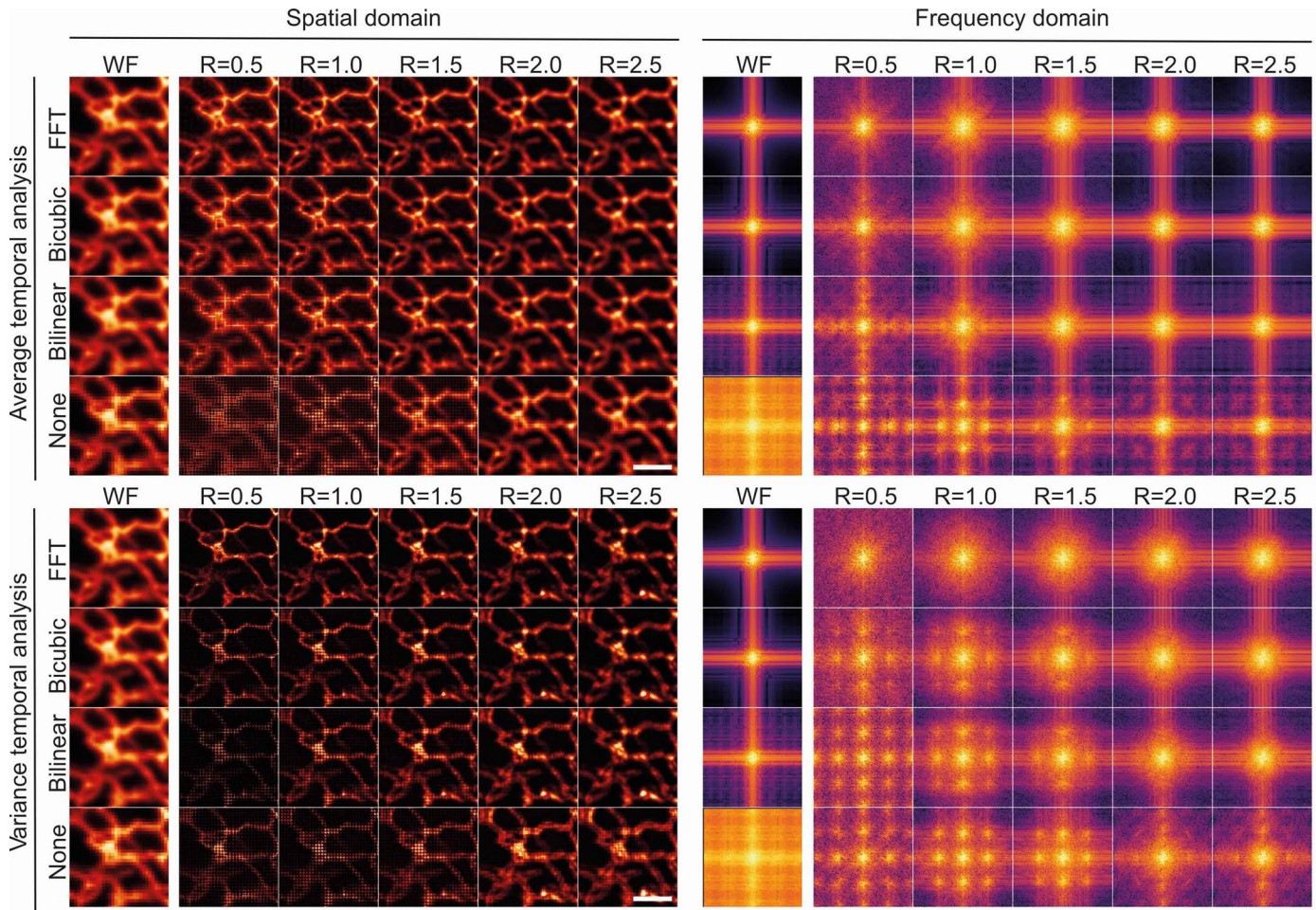

**Extended Data Fig. 1 | Comparison of interpolation methods in the reconstruction of SRRF images.** The presence of macro-pixel artifacts is apparent in both the spatial (left column), and frequency domain (right column) and for AVG (upper row) and VAR temporal analysis reconstruction (lower row) for all interpolation methods apart from the FFT-based interpolation. Data shown corresponds to live COS-7 cells expressing PrSS-mEmerald-KDEL acquired using HiLO-TIRF. Scale bars 2 μm.

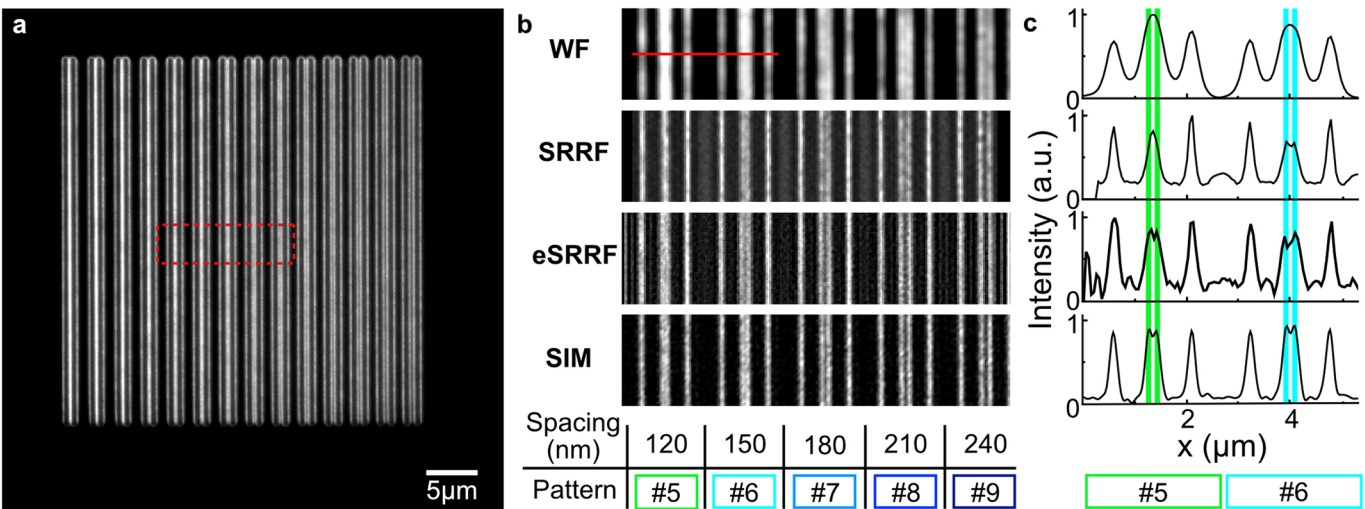

**Extended Data Fig. 2 | Direct comparison of the resolution performance of SRRF, eSRRF and SIM with an ARGO-SIM calibration slide. a** Fluorescence image of the gradually spaced line pattern on the ARGO-SIM calibration slide which displays line pairs with a distance increase of 30 nm per column. **b** Image sections of the area marked by red dashed outline across pattern #5 to #9 (120–240 nm line spacings) of a diffraction-limited (WF) image and the respective super-resolved reconstructions achieved by SRRF (TRA, M = 2, R = 0.6, A = 2, $n_{fr}$ = 5000), eSRRF (AVG, M = 2, R = 1, S = 2, $n_{fr}$ = 5000) and SIM processing, **c** Intensity profiles across pattern #5 and #6 (as indicated by red line in b) allow to resolve the line pair #5 with a distance of 120 nm only in the case of eSRRF and SIM reconstruction, while these lines are not distinguishable in the case of SRRF processing and for the WF data. Scale bar: 5 μm.

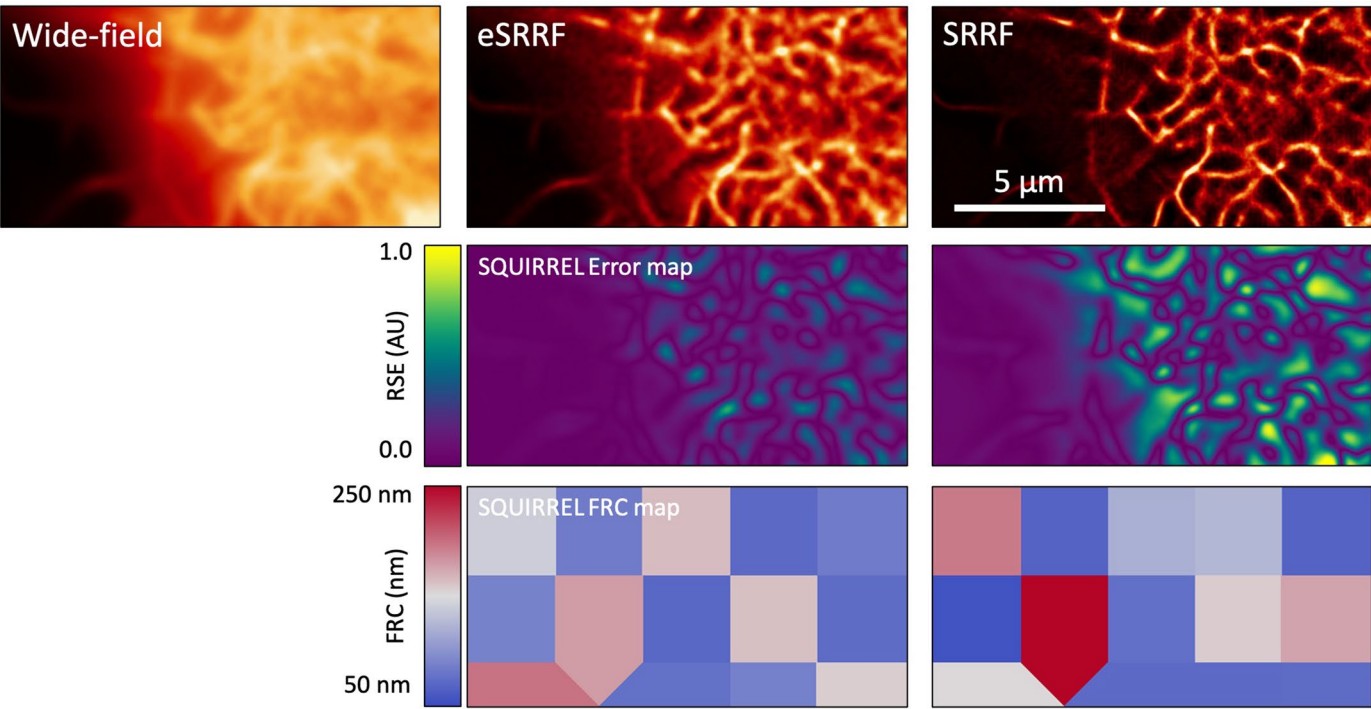

**Extended Data Fig. 3 | SQUIRREL comparison of SRRF and eSRRF.** eSRRF of the actin network (GFP-UtrCH expression, 100 frames at 33 frames/s) in live COS-7 cells shows an improved fidelity with a retained FRC resolution range. The dataset was published before in Culley et al.[21].

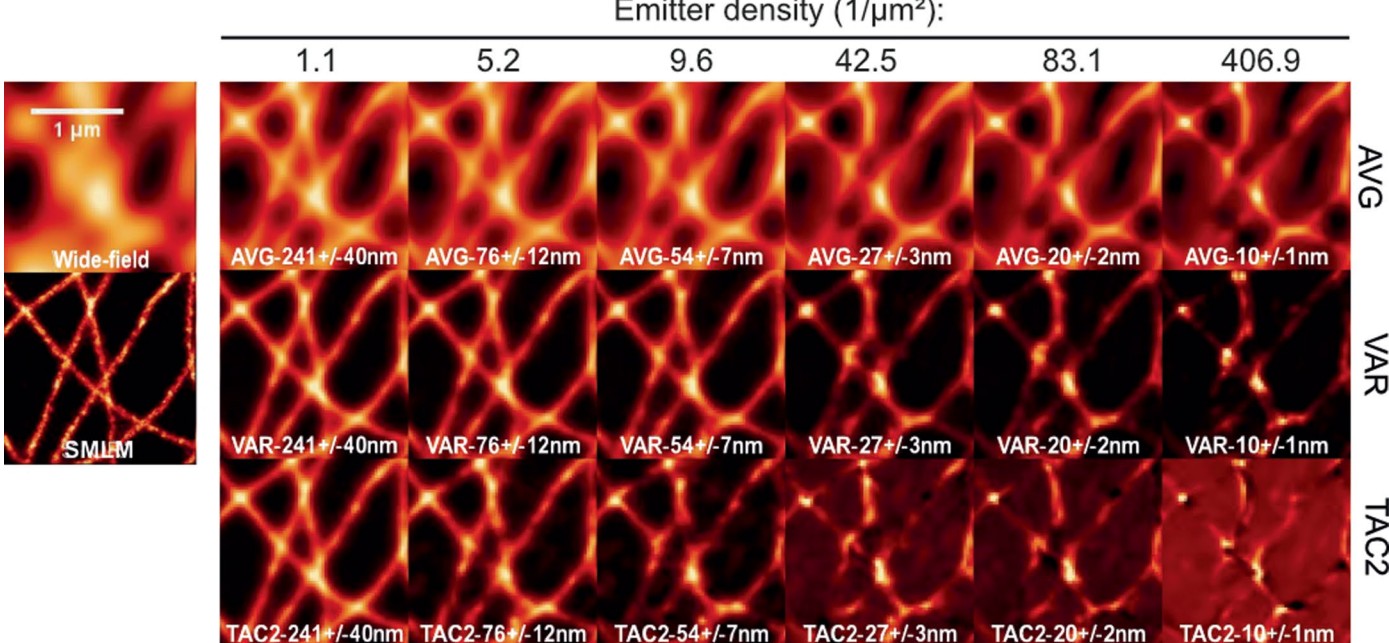

**Extended Data Fig. 4 | Temporal analyses of SMLM data as a function of the density of emitters in the raw data images.** The reconstructed images are shown in increasing emitter density from left to right. The corresponding average nearest-neighbor distance in the binned raw data is stated in each panel. The wide-field and SMLM equivalents are shown on the left for comparison. Scale bar 1 µm.

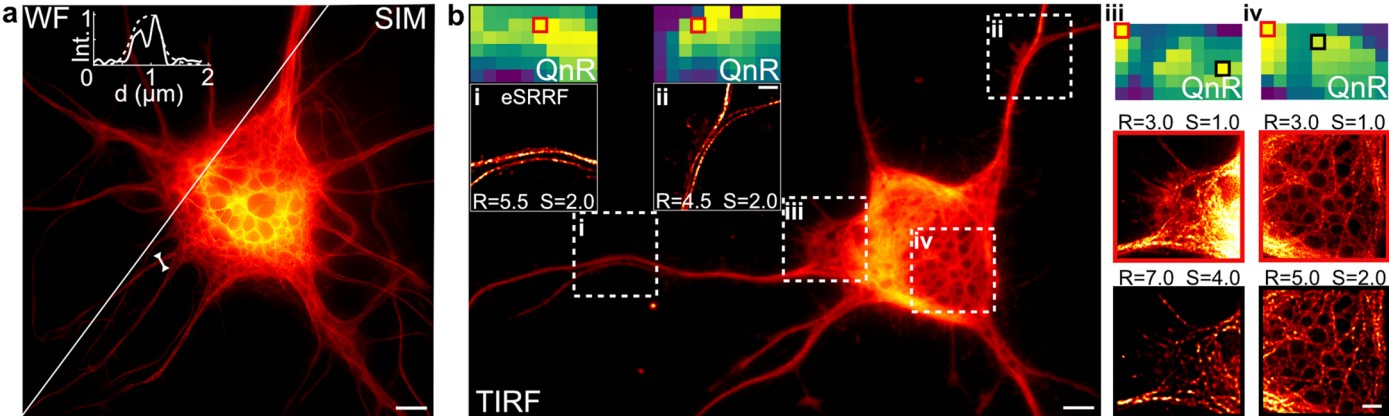

**Extended Data Fig. 5 | Super-resolution imaging of a living cell presenting tubulin bundles within a cultured neuron with SIM and eSRRF. a** The microtubule network in a cultured neuron expressing Tubulin-eGFP can be visualized with SIM at a resolution level (WF-FRC: 253 ± 74 nm, SIM-FRC: 202 ± 47 nm) which allows to resolve microtubule bundles in single dendrites. **b** In cultured neuron expressing Skylan-NS-Tubulin imaged using TIRFM successful eSRRF processing requires user intervention in eSRRF parameter selection due to the local variability and nonlinearity of the QnR map. Especially in large field-of-view images, distinct structural features and variability in marker density result in different optimal parameters for eSRRF super-resolution processing. For example, in this 100 μm × 60 μm image of the microtubule network regions **i** & **ii** display a QnR maximum (marked by the red square) at different positions within the parameter sweep map (left to right: R = 3.0–7.5 in steps of 0.5, top to bottom: S = 1–6 in steps of 1). Furthermore, the QnR map can be nonlinear and especially in the case of pronounced secondary QnR maxima (marked by black squares in **iii** & **iv**) user intervention is required to judge which parameter set yields the best compromise of resolution and fidelity. Scale bar in a&b 5 μm, inset i-iv) scale bars 2 μm.

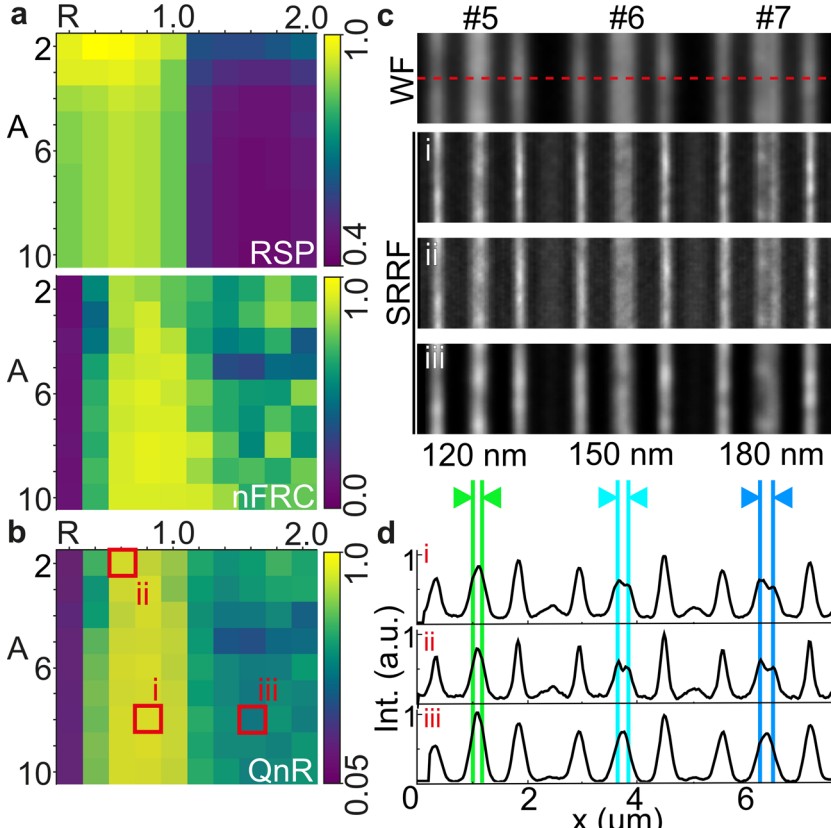

**Extended Data Fig. 6 | Finding optimal parameters for SRRF with the parameter sweep. a** RSP and normalized FRC resolution maps as functions of ring Radius (*R*) and number of Axes in the ring (*A*) reconstruction parameters for SRRF image reconstruction of the gradually spaced line pattern on the ARGO-SIM calibration slide. **b** The compromise between fidelity and FRC resolution is represented in the respective QnR metric map. **c** Wide-field image, optimal SRRF

reconstruction (i, R = 0.8, A = 8, QnR=0.94), reconstruction with high resolution, but with patterning artifacts (ii, R = 0.6, A = 2, QnR=0.89) and low fidelity, low resolution reconstruction (iii, R = 1.6, A = 8, QnR=0.67) of an image section with pattern #5 (120 nm), #6 (150 nm) and #7 (180 nm). **d** The line profiles across the SRRF reconstructions i, ii, and iii show that pattern #5 can't be resolved by any of the parameter sets.

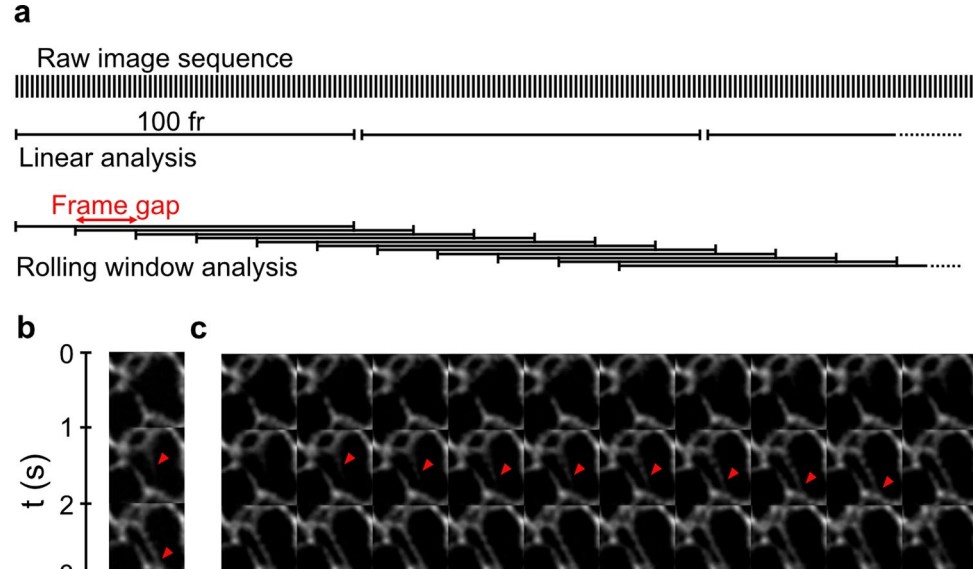

**Extended Data Fig. 7 | Increasing temporal sampling by rolling window analysis. a** Super-resolved temporal image stack is reconstructed by analyzing consecutive frame windows (linear analysis). A rolling window analysis with a frame gap of less than the window size, similar to what has been implemented for SIM[80,81], allows to increase the temporal sampling rate and can translate into a higher temporal resolution without sacrificing spatial resolution. **b** This allows to visualize dynamic rearrangement of the endoplasmic reticulum (ER, red arrow) acquired by live-cell HiLO-TIRF of COS-7 cells expressing PrSS-mEmerald-KDEL at a frame rate of 1 Hz with a 100 frame-window linear eSRRF analysis. **c** The rolling window analysis with a frame gap of 10 frame allows to increase the temporal sampling to 10 Hz, thus, revealing the substeps of the ER tubule formation or rearrangement (red arrow), image section width 3 μm, FRC resolution (mean ± standard deviation): eSRRF: 143 ± 56 nm, Scale bar 2 μm.

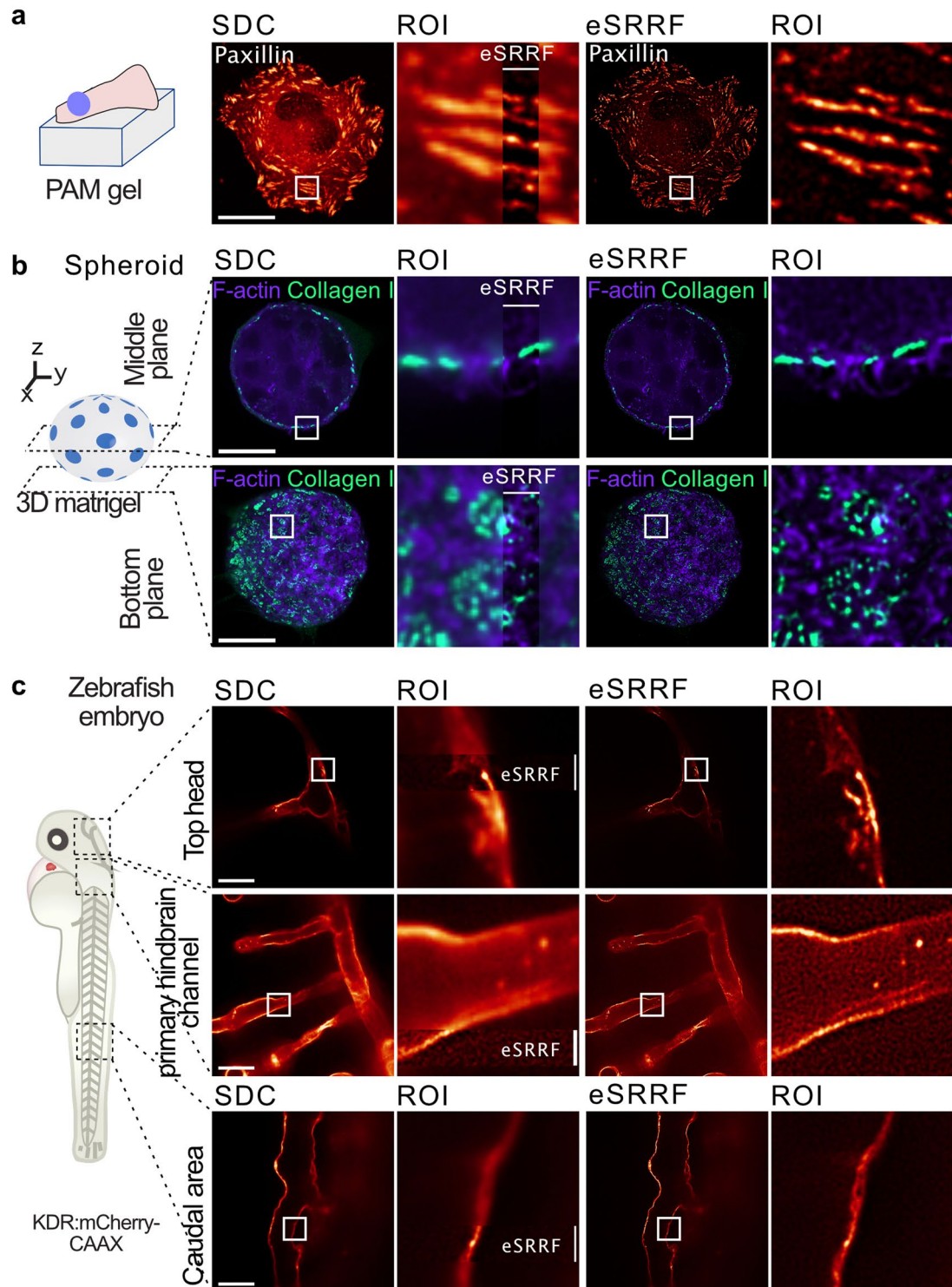

**Extended Data Fig. 8 | eSRRF enhances spinning-disk confocal imaging deep inside fixed and live cells, spheroid and organisms. a** U2OS cells expressing endogenously tagged paxillin were plated on 9.6 kPa polyacrylamide (PAM) gels and were imaged live using a spinning-disk confocal (SDC, resolution estimate: FRC = 573.0 ± 7.2 nm) and processed with eSRRF (resolution estimate: FRC = 197 ± 34 nm). **b** DCIS.com lifeact-RFP cells forming a spheroid in 3D matrigel in the presence of fluorescently labeled collagen I. Samples were fixed and imaged using a spinning-disk confocal microscope (resolution estimate: FRC(Actin)= 569 ± 59 nm/FRC(Collagen I) = 583 ± 14 nm) and processed using eSRRF (resolution estimate: FRC(Actin)= 229 ± 97 nm/FRC(Collagen I) = 130 ± 36 nm). Representative fields of view highlighting the spheroids' middle and bottom planes are displayed. **c** Zebrafish embryos expressing mcherryCAAX in the endothelium were imaged live using a spinning-disk confocal. Vessels located at different parts of the embryo were imaged. For all panels, the eSRRF reconstructed images (resolution estimate: FRC(top) = 194 ± 37 nm/FRC(middle) = 393 ± 23 nm/FRC(bottom)= 307 ± 60 nm) and the original spinning disk images are displayed (resolution estimate: FRC(top) = 641 ± 82 nm/FRC(middle)= 573.9 ± 5.7 nm/FRC(bottom)=575.9 ± 7.9 nm). The ROI zoom-ins are marked by white squares and the line profile position is marked in the eSRRF zoom-in by a white line. Intensity profiles of the diffraction-limited SDC data (dashed line) and eSRRF reconstruction are plotted in the panel on the very right. Scale bars 25 μm.

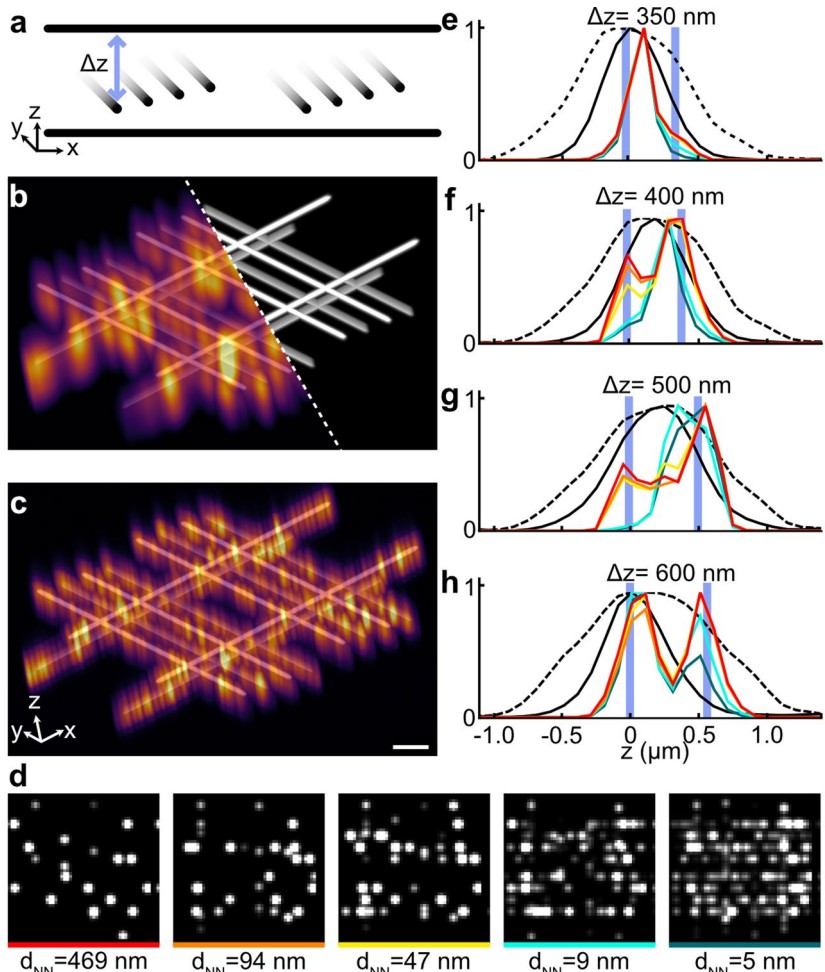

**Extended Data Fig. 9 | 3D eSRRF on a simulated woodpile structure dataset at different emitter densities. a** A woodpile structure with filaments along the x- and y-axis with varying axial distances Δz served as a ground truth 3D architecture. **b** From a simulated fluorescence imaging stacks at nine focal planes based on the ground truth architecture (white lines) a 3D view as directly observed in a MFM can be rendered. **c** 3D eSRRF processing of the simulated datasets allows to enhance the lateral and axial resolution. **d** Representative image frames with a width of 5 μm of simulated datasets generated at five different emitter densities ranging over two orders of magnitude from an average next neighbor distance $d_{NN}$ of 469 nm to 5 nm. **e-h** Axial profiles of filaments crossing at a distance Δz of 350 nm (**e**), 400 nm (**f**), 500 nm (**g**) and 600 nm (**h**) for a sum of all simulated images (black dashed line), a deconvolved 3D reconstruction (black solid line) or eSRRF reconstructions of image stacks with very low (red), low (orange), medium (yellow), high (cyan) and very high (green) emitter densities. Scale bar 500 nm.

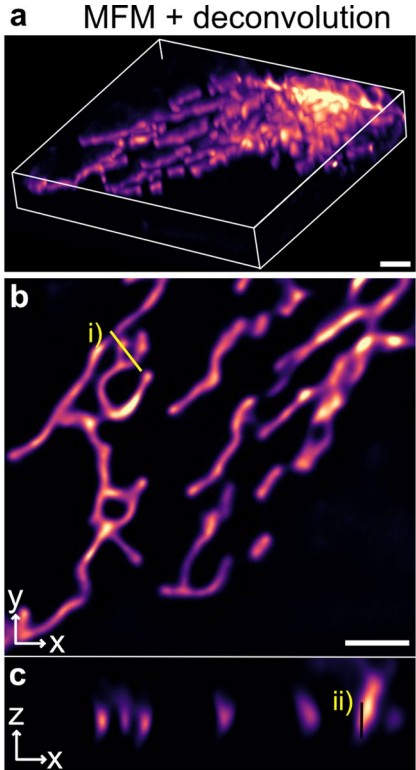

**a** MFM + deconvolution

**Extended Data Fig. 10 | Deconvolved 3D image stack of U2OS cells expressing TOM20-Halo, loaded with JF549 acquired with MFM. a** 3D rendering, **b** single *z*-slice and **c** single cropped y-slice. i) and ii) are the line profiles in *x*,*y* and *z*-plane displayed in Fig. 4. Scale bars 2 μm.

# nature research

|  |  |
|---|---|

# Reporting Summary

Nature Research wishes to improve the reproducibility of the work that we publish. This form provides structure for consistency and transparency in reporting. For further information on Nature Research policies, see our Editorial Policies and the Editorial Policy Checklist.

## Statistics

For all statistical analyses, confirm that the following items are present in the figure legend, table legend, main text, or Methods section.

| n/a | Confirmed |  |
|---|---|---|
| ☐ | ☒ | The exact sample size (*n*) for each experimental group/condition, given as a discrete number and unit of measurement |
| ☒ | ☐ | A statement on whether measurements were taken from distinct samples or whether the same sample was measured repeatedly |
| ☒ | ☐ | The statistical test(s) used AND whether they are one- or two-sided <br> *Only common tests should be described solely by name; describe more complex techniques in the Methods section.* |
| ☒ | ☐ | A description of all covariates tested |
| ☒ | ☐ | A description of any assumptions or corrections, such as tests of normality and adjustment for multiple comparisons |
| ☐ | ☒ | A full description of the statistical parameters including central tendency (e.g. means) or other basic estimates (e.g. regression coefficient) AND variation (e.g. standard deviation) or associated estimates of uncertainty (e.g. confidence intervals) |
| ☒ | ☐ | For null hypothesis testing, the test statistic (e.g. *F*, *t*, *r*) with confidence intervals, effect sizes, degrees of freedom and *P* value noted <br> *Give P values as exact values whenever suitable.* |
| ☒ | ☐ | For Bayesian analysis, information on the choice of priors and Markov chain Monte Carlo settings |
| ☒ | ☐ | For hierarchical and complex designs, identification of the appropriate level for tests and full reporting of outcomes |
| ☐ | ☒ | Estimates of effect sizes (e.g. Cohen's *d*, Pearson's *r*), indicating how they were calculated |

*Our web collection on statistics for biologists contains articles on many of the points above.*

## Software and code

Policy information about availability of computer code

| Data collection | Commercial: Acquisition with Nikon-NIS Elements 5.30.02/5.30.05 (Nikon), SlideBook V6 (Intelligent Imaging Innovations, Inc.), ZEN Black Version 11.0.2.190 (Carl Zeiss) <br> Open Source: Fiji/ImageJ 1.45f (plugins: NanoJ-eSRRF) <br> Custom code: Diffusing particle simlation: Python script available as GoogleCoLabs Jupyter notebook on GitHub: https://github.com/HenriquesLab/NanoJ-eSRRF. |
|---|---|
| Data analysis | Commercial: Huygens Professional version 21.10 (Scientific Volume Imaging, The Netherlands, http.//svi.nl) <br> Open Source: Fiji/ImageJ 1.45f (plugins: NanoJ-eSRRF, ThunderSTORM, NanoJ-SQUIRREL, LLSM, HAWK, DeconvolutionLab2), devbio-Napari (https://github.com/haesleinhuepf/devbio-napari) |

For manuscripts utilizing custom algorithms or software that are central to the research but not yet described in published literature, software must be made available to editors and reviewers. We strongly encourage code deposition in a community repository (e.g. GitHub). See the Nature Research guidelines for submitting code & software for further information.

## Data

Policy information about availability of data

All manuscripts must include a data availability statement. This statement should provide the following information, where applicable:
- Accession codes, unique identifiers, or web links for publicly available datasets
- A list of figures that have associated raw data
- A description of any restrictions on data availability

Datasets are available on Zeondo (https://doi.org/10.5281/zenodo.6466472).

# Field-specific reporting

Please select the one below that is the best fit for your research. If you are not sure, read the appropriate sections before making your selection.

☒ Life sciences ☐ Behavioural & social sciences ☐ Ecological, evolutionary & environmental sciences

For a reference copy of the document with all sections, see nature.com/documents/nr-reporting-summary-flat.pdf

# Life sciences study design

All studies must disclose on these points even when the disclosure is negative.

| Sample size | No sample-size calculation was performed as the manuscript reports the demonstration of an image reconstruction method, we chose the sample size that can validate reproducibility of our technique. The manuscript draws no biological conclusions, and does not examine or compare different biological conditions. This is not a life science study with comparative analyses of a certain sample size. |
|---|---|
| Data exclusions | No data was excluded from the analysis |
| Replication | All attempts of replication were successful and results supported by simulations. All experiments were repeated three or more times with similar results. |
| Randomization | Randomization is not relevant to this study as this is not a life science study with comparative analyses of biological situations, but a validation of the performance of an algorithm supported by a data driven parameter optimization to avoid user bias. |
| Blinding | No blinding was performed. Blinding is not relevant because there is no comparison of different biological situations performed in this work. |

# Reporting for specific materials, systems and methods

We require information from authors about some types of materials, experimental systems and methods used in many studies. Here, indicate whether each material, system or method listed is relevant to your study. If you are not sure if a list item applies to your research, read the appropriate section before selecting a response.

### Materials & experimental systems

| n/a | Involved in the study |
|---|---|
| ☐ | ☒ Antibodies |
| ☐ | ☒ Eukaryotic cell lines |
| ☒ | ☐ Palaeontology and archaeology |
| ☐ | ☒ Animals and other organisms |
| ☒ | ☐ Human research participants |
| ☒ | ☐ Clinical data |
| ☒ | ☐ Dual use research of concern |

### Methods

| n/a | Involved in the study |
|---|---|
| ☒ | ☐ ChIP-seq |
| ☒ | ☐ Flow cytometry |
| ☒ | ☐ MRI-based neuroimaging |

## Antibodies

| Antibodies used | anti-α-tubulin mouse monoclonal IgG1 antibodies (DM1A (T6199, Sigma) and B-5–1-2 (T5168, Sigma)
goat anti-mouse antibody conjugated to a DNA sequence (P1 docking strand, U10001, Ultivue kit, discontinued) |
|---|---|
| Validation | DM1A: The isotype is determined by a double diffusion immunoassay using Mouse Monoclonal Antibody Isotyping Reagents, Product Number ISO2. Anti-a-Tubulin antibody, Mouse monoclonal recognizes an epitope located at the C-terminal end of the a-tubulin isoform (amino acids 426-430) in a variety of organisms (e.g., human, bovine, mouse, and chicken).

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

# Eukaryotic cell lines

Policy information about cell lines

| | |
|---|---|
| Cell line source(s) | COS-7 cells (ATCC CRL-1651),  HeLa cells (ATCC CRM-CCL-2), Jurkat (Cellbank Australia Jurkat-ILA1), U2OS (ATCC HTB-96 & Leibniz Institute DSMZ-German Collection of Microorganisms and Cell Cultures, Braunschweig DE, ACC 785), MCF10 DCIS.COM (DCIS.COM) |
| Authentication | Cell lines from ATCC, DSMZ, DCIS.COM or Cellbank Australia used at low passage numbers and authenticated with a morphology check under the light microscope. |
| Mycoplasma contamination | All cell lines were tested negative for mycoplasma contamination on a regular basis. |
| Commonly misidentified lines (See ICLAC register) | No commonly misidentified cell lines were used in the study. |

# Animals and other organisms

Policy information about studies involving animals; ARRIVE guidelines recommended for reporting animal research

| | |
|---|---|
| Laboratory animals | Housing and experimentation of Zebrafish (Danio rerio, strain Tg(KDR:mcherryCAAX, age 2 days post fertilization, sex not not determinable) were performed under license MMM/465/712-93 (issued by the Ministry of Agriculture and Forestry, Finland). Rat neuronal cultures were prepared from Wistar rat day 18 embryos of both sexes in agreement with the guidelines established by the European Animal Care and Use Committee (86/609/CEE) and was approved by the Aix-Marseille University ethics committee 14 (France, agreement G13O555) |
| Wild animals | The study did not involve wild animals. |
| Field-collected samples | The study did not involve field-collected samples. |
| Ethics oversight | Zebrafish handling was performed under license MMM/465/712-93 issued by the Ministry of Agriculture and Forestry, Finland. Rat neuronal culturing was performed under license G13O555 issued by the Aix-Marseille University ethics committee 14 (France). |

Note that full information on the approval of the study protocol must also be provided in the manuscript.

