## [Peer Review File · Nature Methods]

Peer Review Information

Manuscript Title: High-fidelity 3D live-cell nanoscopy through data-driven enhanced super-resolution radial fluctuation

Corresponding author name(s): Ricardo Henriques, Bassam Hajj , Christophe Leterrier

Editorial Notes: n/a

Reviewer Comments & Decisions:

Decision Letter, initial version:

Dear Ricardo,

Your Article entitled "High-fidelity 3D live-cell nanoscopy through data-driven enhanced super-resolution radial fluctuation" has now been seen by two reviewers, whose comments are attached. While they find your work of some potential interest, they have raised concerns which in our view are sufficiently important that they preclude publication of the work in Nature Methods.

We will consider looking at a revised manuscript only if further experimental data allow you to address all the major criticisms of the reviewers (unless, of course, something similar has by then been accepted at Nature Methods or appeared elsewhere). This includes submission or publication of a portion of this work somewhere else.

We are not overly concerned with issues of 'novelty', but we are concerned that the reviewers were not convinced that the work represents an important practical advance over SRRF based tools developed by your group and others. We would only be interested in seeing a revised version where these benefits are more clearly delineated and the other technical concerns are addressed. We hope you understand that until we have read the revised paper in its entirety we cannot promise that it will be sent back for peer-review.

If you are interested in revising this manuscript for submission to Nature Methods in the future, please contact me to discuss your appeal before making any revisions. Otherwise, we hope that you find the reviewers' comments helpful when preparing your paper for submission elsewhere.

Sincerely,
Rita

Rita Strack, Ph.D.
Senior Editor
Nature Methods

Reviewers' Comments:

Reviewer #1:
[Attached]

This paper presents a new software tool eSRRF for achieving super-resolution from fluctuations of fluorescence intensity. The paper is well written and presented in an appealing manner. However, there are several major reasons, which makes me think that this work is not a good match to Nature Methods. While the major concerns below are the reasons for my conclusion, I additionally present several other significant concerns that need attention of the authors and some minor comments regarding certain details.

Major:

Line 107: "In eSRRF, we redefined ... reconstructions". In my observation, there is no redefinition of fundamental principles. Radiality now uses interpolation in Fourier transform, and temporal analysis (for window selection) uses SSIM plots. These are in my opinion, well-known ideas/approaches and there is no fundamental principle redefinition involved. They are rather trivial tricks to improve interpolation quality and guide the selection of parameter, respectively. Similarly, neither FRC nor RSP are new (they have been used in SQUIRREL already).

It appears that this work is simply combining the two previous works of the authors – some concepts of SRRF and some of SQUIRREL – and programming them into one pipeline for searching an optimal candidate in the parameter space. None of the involved ideas are new or groundbreaking, or an important new method. At best, it is an incremental contribution and essentially just a more elaborate post-processing than before. This is also accepted by the authors in Lines 140-142 ("The increased image fidelity results from the implementation of several new and optimized routines in the radial fluctuation analysis algorithm, introduced through a full rewriting of the code.").

Indeed, the approach of eSRRF is more computationally demanding as now the SRRF images need to be first constructed using each candidate in the parameter space (exhaustive search for the best set of parameters). Where the work is devoid generally of any new ideas over their previous works, it is strange that the authors neglect an obvious observation that their FRC and RSP maps both appear continuous, and an exhaustive search is not needed. I therefore rank the work quite low in any form of novelty over their previous works.

3D super-resolution: The authors claim 3D resolution improvement by applying eSRRF on fluctuations data acquired using MFM. The results in Fig. 4 show only a contrast enhancement over diffraction limited image. The dip between the structures is clearly visible in numerical sense in the diffraction limited image, and the eSRRF results only enhance the contrast by pronouncing the dip between the structures. Indeed, the result is better than the result of deconvolution (no details about how this was done are available), and this says only that the deconvolution was not applied well or was not effective. Similarly, in sup. Fig. 11(a-c), it appears again that eSRRF accomplishes only contrast enhancement.

Significant:

Parameters: The ranges of radius and sensitivity seems like still some parameters that the user may need to specify and need insight of. It appears that sensitivity is just some form of non-linearity. Clarification is needed on this parameter. It does not appear in the original SRRF article from 2016.

Please see line 97: "However, obtaining optimal reconstruction results ... from reconstruction artifacts and lack signal linearity." Also, line 138. These are strong statements. It will be useful to refer to some results/articles that discuss the reconstruction artifacts in fluctuation based techniques.

Use of FRC as the resolution metric: FRC presents some challenges, as has been reported in the literature. It is highly recommended that the authors show resolution using at least one more approach. The limitation of FRC as a resolution measurement for eSRRF is evident in sup. Fig. 7. As the

results in this figure show, FRC is generally an over-optimistic resolution estimate for eSRRF. It is recommended that authors back the resolution claims using decorrelation based metric [<https://www.nature.com/articles/s41592-019-0515-7>] in addition to FRC. For the same reason, I suggest that the resolution number reported in the graphical abstract be removed.

The use of RSP as a fidelity measure and nFRC as resolution metric: It is not clear what exactly is the RSP measuring that the authors refer to as fidelity. Consider in sup. Mov. 1. When $S=2$, $R=2$, then there is a lot of debris in the background. Neither do the foreground structures appear in high fidelity. But the RSP score is quite high. So, it does not seem that RSP is performing as a fidelity metric. The normalization of FRC is not clearly discussed. Does $nFRC=0$ corresponds to the lowest value of FRC in the search space and $nFRC=1$ corresponds to the highest? What if the search space is not optimally selected in the first place? Isn't it more appropriate to set $nFRC=0$ as the FRC related to the diffraction limited image?

Benchmark for artefact suppression and ground truth: In context of several figures, but more specifically sup. Fig. 4, eSRRF presents poorer contrast and presence of more details than SRRF. It is not clear if the more details in eSRRF are artifacts or whether the SRRF has suppressed these details. Data driven error analysis such as in SQUIRREL are useful. But artefacts are better studied using SEM images or other more precise and accurate imaging modalities. In this sense, the precision of STORM images is such that resolution +/- precision of them is quite close to eSRRF. In this case, using even STORM as benchmark is not useful. In the same vein for sup. Fig. 5, it does not make sense to compute the average nearest neighbor distance below the precision of STORM images. Therefore, I do not trust the numbers reported in the last 2, even the last 3 columns of this figure. In sup. Fig. 11, it is difficult to say that what is reconstructed is reliable. The question is quite important for thick samples considered in this figure, since the thick samples contribute large scattering and the details reconstructed can easily be inaccurate.

Rolling window: In sup. Fig. 10, two consecutive time windows have 90 frames in common. How do we pin the time of ER formation in this case? How do we know that it is about formation and not coming into the imaging plane from out-of-focus? Also, there are quite a few discrepancies in this figure. In the caption, the temporal sampling is reported as 10 Hz, whereas in the main text, it is reported as 50 ms (i.e. 20 Hz). Image section width is reported as 3 μm , whereas the scale bar is shown as 2 μm (which is visibly more than 2/3rd of the image section). I suggest that the authors check these details.

Minor:

Line 168: how was the temporal binning performed. Was it binning consecutive frames together or in an interleaved fashion? In the first case, photobleaching introduces sparsity in the later frames. In the second case, the signal to noise ratio is relatively uniform across the binned frames.

The authors just coin the term QnR, not providing any elaboration of it. Is it an acronym? What does it stand for?

In Fig. 3, the use of NSTORM in (a) and Fast-GI TIRF in (b) is quite confusing. Either it is incorrect annotation, or this annotation was done for some other purpose, or the images need to be checked.

In sup. Tab. 3, it will be useful to include optical parameters of the systems, diffraction limited resolution, SRRF resolution, and eSRRF resolution in each case.

In the simulated examples, more details are needed. In sup. Fig. 3, what is the distance between emitters (seems larger than the claimed resolution) in the third circle (where emitters are not resolved by eSRRF). What is the distance between the circles?

In sup. Fig. 11b, Collagen and F-actin seems color coded, but inclusion of them as different texts in the two planes seems misleading. The upper plan has text F-Actin written on it and the lower plan has Collagen. However, it seems both the planes show both collagen and F-actin.

In sup. Fig. 12, it would have been useful to indicate the plane in which the different lines lie, so that appreciation of the sub-diffraction distance between the different lines is easier.

Reviewer #3:

Remarks to the Author:

See review attachments as Figure is included.

[Attached]

I read the eSRRF manuscript (including the corresponding bioRxiv preprint version) by Professor Henriques and co-workers with great interest. The main novel developments include: 1) Updated SRRF algorithm plus adaptive parameters (radius, sensitivity, number of frames) choosing for high-fidelity or balanced fidelity-resolution reconstruction; 2) 3D SRRF realization by extending the previous SRRF theory to axial dimension and incorporating multi-focus imaging. However, regrettably, although the manuscript does have some novelty and contribution, I am afraid it is not sufficient and thus couldn't recommend its publication on high-quality journals including Nature Methods.

- 1) After it was first proposed by Henriques' group, SRRF method, including the corresponding algorithm, has been followed and improved by several groups. However, the current manuscript neither includes any literature review in the Introduction, nor performs any simulation or experimental comparison between the proposed eSRRF algorithm (under the situation without adaptable parameter searching) and the other developments.
- 2) Did the authors implement the parameter exploration scheme into their original SRRF reconstruction? I would like to see the comparison among SRRF reconstruction with and without QnR parameter searching, and also eSRRF reconstruction with and without QnR parameter searching, which will help distinguish which steps improve the resolution or fidelity.
- 3) How to evaluate the parameters based on automated QnR searching is "optimal" or not? This is different from the optimal PSF design in 3D SMLM based on CRLB metric.
- 4) I agree that high-fidelity imaging is more and more urgent with the super-fast development of super-resolution imaging field. However, the strategy of automated reconstruction parameter search and finetune based on some metrics to improve image fidelity has been already used in many fields, including the super-resolution imaging. Typically, neural network training with the feedback of loss functions.
- 5) The 3D SRRF has been already achieved by another group [1,2].
- 6) The authors demonstrate the estimation of the number of frames necessary for SRRF reconstruction based on tSSIM metric. However, this can only be achieved in live-cell experiments, how to determine the frame parameter in general fix-cell imaging?
- 7) The section "eSRRF works across a wide range of live-cell imaging modalities" doesn't provide any new information as the universal characteristic of SRRF reconstruction applied to different modalities has already been shown, including their first Nature Communications SRRF paper.
- 8) The authors claimed "live-cell SDC and LLS imaging" in the text, however, there is no time-lapse data shown in the figure or video (only time-lapse TIRF-eSRRF data). In my understanding, SRRF will hugely increase the acquisition time in 3D scanning as hundreds to thousands of frames need to be acquired in each axial plane.

- 9) The explanation about the implementation of MFM into SRRF is too little to be understood well. How is this achieved experimentally? What is the hardware system like? This is one of the main contribution of the manuscript and thus more details should be shown.
- 10) The reconstructed depth of MFM-SRRF is relatively shallow, 3.6 μm . What's the limit?
- 11) 3D live mitochondria experiments. The comparison between the eSRRF (Figure 4) and deconvolution (Supplementary Fig. 14) results confuse me a lot. The selected xy (i) and xz (ii) intensity profiles are too subjective and unconvincing (including other figures, such as Figure 3, Supplementary Fig.2.). In my view, eSRRF and deconv results are very similar and can't provide me with obvious difference. Actually, I even think deconv result performs better.
- 12) Movie S4. Within the displayed short imaging duration of 17s, it seems that the photobleaching and phototoxicity are pretty severe as the whole mitochondria network contracts quickly. The mitochondria also become fewer and fewer and more fragmented. This concerns me about the practicality of this method.
- 13) The temporal resolution and duration of (3D) (e)SRRF shown in the manuscript is too low considering the rapid development of the ultrafast and long-term super-resolution imaging filed, either in 2D or 3D, such as Hessian SIM, GI SIM, AiryScan, IDDR-SPIM, 3D pRESOLFT, etc., although one may argue that SRRF has some advantages.
- 14) The reconstructed results confused me often. In some cases, it seems that SRRF images have better resolution in spite of the fidelity (following figure (a, c, d)), while in other cases, eSRRF outperforms (following figure (b)).

- 1) RSP metric should be explained at its first appearance.
- 2) Scale bar is missed in Fig.1
- 3) Fig. 1b. Conventional SRRF reconstruction steps should be added to the figure for better comparison.
- 4) Fig. 1c. FRC resolution of SMLM is 71nm. Why the DNA-PAINT resolution is so low, which doesn't matches my experimental experience.
- 5) Fig. 2d-e are missed (preprint version has).
- 6) Supplementary Movie 4. The note should be "eSRRF" rather than "SRRF". Besides, the same period is repeated several times.

[1] Chen, Rong, et al. "Efficient super-resolution volumetric imaging by radial fluctuation Bayesian analysis light-sheet microscopy." *Journal of biophotonics* 13.8 (2020): e201960242.

[2] Zhao, Yuxuan, et al. "Isotropic super-resolution light-sheet microscopy of dynamic intracellular structures at subsecond timescales." *Nature Methods* 19.3 (2022): 359-369.

Author Rebuttal to Initial comments

Dear Reviewers and Editorial Team at Nature Methods,

We are grateful for your time and effort in reviewing our manuscript entitled "High-fidelity 3D live-cell nanoscopy through data-driven enhanced super-resolution radial fluctuation". Your insightful feedback and constructive criticism have been instrumental in improving the quality of our work.

In response to your comments, we have carefully revised the manuscript and made several significant changes, which we believe have addressed the points you raised. Specifically, we have added two new Supplementary Figures that demonstrate the impact of algorithm refinement and parameter optimization using an Argo-SIM calibration structure (Supplementary Figure 3&10). Additionally, we have included new datasets acquired with SkylanS, a fluorescent protein optimized for fluorescence fluctuation analysis, to demonstrate super-resolution imaging in cultured neurons (Figure 3a) and the observation of dynamic actin rearrangement in U2OS cells over 12 hours at super-resolution level (Figure 3e & Supplementary Movie 4).

To provide better insight into the implementation and performance of 3D eSRRF, we have added a comprehensive description of the principles and practical implementations of extending eSRRF to 3D in the new Supplementary Note 3. Furthermore, we have included new simulated data to showcase the performance and limitations of the axial super-resolution capabilities (Supplementary Figure 15), and experimental MFM data of volumetric super-resolution imaging for more than three minutes without compromising cell health (Supplementary Movie 5).

We have also extended the introduction and discussion part of the manuscript to provide more context and comparison with previously published live-cell super-resolution imaging approaches, both in 2D and 3D (Supplementary Table 2 and 5).

We believe that these revisions have significantly strengthened the paper, and we hope that you find the revised manuscript satisfactory. We appreciate your continued support, and we welcome any additional feedback or comments you may have on this new version.

Once again, we thank you for your valuable feedback and for your time in reviewing our manuscript.

All changes in the manuscript & supplementary material and answers to the reviewers are marked in blue.

Reviewer comments received July 13th 2022

Reviewer 1:

This paper presents a new software tool eSRRF for achieving super-resolution from fluctuations of fluorescence intensity. The paper is well written and presented in an appealing manner. However, there are several major reasons, which makes me think that this work is not a good match to Nature Methods. While the major concerns below are the reasons for my conclusion, I additionally present several other significant concerns that need attention of the authors and some minor comments regarding certain details.

Major:

- Line 107: "In eSRRF, we redefined ... reconstructions". In my observation, there is no redefinition of fundamental principles. Radiality now uses interpolation in Fourier transform, and temporal analysis (for window selection) uses SSIM plots. These are in my opinion, well-known ideas/approaches and there is no fundamental principle redefinition involved. They are rather trivial tricks to improve interpolation quality and guide the selection of parameter, respectively.

We thank the reviewer for their detailed overview of the manuscript. Regarding the point of "no fundamental principle redefinition involved", we would like to highlight that this is the first demonstration of a super-resolution approach where both resolution and quality (artifact ranking) metrics are combined in a data-driven manner to self-optimize the algorithm used, thus producing a super-resolution approach that generates images of better combined resolution and quality than previously possible. We now further strengthen this point in the new supplementary figure 3, where it is evidenced that the novel mathematical basis developed for eSRRF allows it to achieve a better resolving capacity than SRRF, even when SRRF parameters are optimized. We agree with the reviewer that we have used many best-in-class known mathematical approaches to fine-tune the algorithm. The same can be argued regarding most gold standard algorithms in the super-resolution field, for example - FRC itself (Nieuwenhuizen et al., *N. Meth.*, 2013), whereby the FRC approach was widely used in the cryo-EM field prior to being adapted for super-resolution microscopy. It is important to recognize that the unique combination of the mathematical approaches we employ allows us to propose new concepts, such as the newly created QnR maps, whose quantitative analysis is at the root of the data-driven optimization proposed in eSRRF. QnR maps also solve a major reproducibility problem in the field by guiding users in the selection of parameters, rather than leaving their selection as an arbitrary choice.

To further substantiate the novelty of the QnR maps, consider that despite the measures of quality (RSP) and resolution (FRC) being gold standards in the field, they have not been used in this context of optimizing reconstruction parameters for the best compromise between image fidelity and resolution before. Other works have only focused on assessing the two aspects separately, for example: Schäfer et al., *Journal of Microscopy* (2004), Ball et al., *Scientific Reports* (2015), Zhao et al., *bioRxiv* (2022), <https://doi.org/10.1101/2022.12.12.520072>). Subjective, user-defined image quality scores have also been used in other studies (Durand et al., *Nat Comm* (2018)). To our knowledge, there is only one approach that considers both image quality and resolution by employing a multicomponent loss function to train an ESRGAN for deep learning based super-resolution image reconstruction (Chen et al., *bioRxiv* (2021), <https://doi.org/10.1101/2021.10.08.463746>). However, the two last concepts mentioned (proposed by Durand et al. and Chen et al.) are deep learning approaches that optimize the output quality based on the concept of a loss function dependent on prior assumptions. In contrast, the data-driven eSRRF parameter optimization does not depend on a pretrained model, prior knowledge about the dataset, simulated ground truth, or annotated data, basing its scoring on the direct unsupervised analysis of the data being collected via FRC and RSP calculations. This procedure is both simpler and more robust when the data being collected has new features or properties not observed before. Furthermore, the implemented parameter optimization based on the QnR metric directly provides insight into quantitative and

comprehensible measures of image fidelity and resolution, enabling the user to critically analyze the results and take an informed decision.

We have added the following section to the discussion section of the revised version of the manuscript (lines 424-434):

"While the performance of eSRRF may be surpassed in spatial or temporal resolution by deep-learning based SRM approaches, these face a very specific set of limitations. Such deep-learning methods have been shown to convert sparse SMLM data⁶⁰ or even diffraction-limited images of dynamic structures⁶¹ into super-resolved time series at rates above 50 Hz. Their extension to 3D isotropic volumetric live-cell super-resolution microscopy has demonstrated imaging rates of up to 17 Hz⁵⁸. However, unlike eSRRF, these methods require prior knowledge of the dataset and generally require either simulated or experimental ground truth data. These features also mean that insufficient, unbalanced, or unsuitable data can lead to severe image degradation and hallucinations by deep-learning methods, that are not easy to detect.⁶²

Here, we have introduced a data-driven parameter optimization approach that aids users in selecting optimal parameters learned directly from the data to be analyzed. These optimal parameters are chosen by balancing the need for high reconstruction fidelity together with high spatial and temporal resolution. In contrast to the training procedures employed in deep learning approaches, the data-driven eSRRF parameter optimization bases its scoring on the direct unsupervised analysis of the data being collected via FRC and RSP calculations. As such, an eSRRF analysis is easy and reliable, especially when the data being collected has new properties or features that haven't been seen before. Furthermore, the implemented parameter optimization based on the QnR metric directly provides dataset specific insight into the relationship between image fidelity and resolution, enabling the user to critically analyze results and take an informed decision on analysis settings. By combining the QnR-based parameter sweep with a temporal window optimization, eSRRF achieves optimal spatial and temporal resolution while minimizing reconstruction artifacts and reducing user bias. This makes eSRRF a super-resolution method that shows users how to best analyze their data and gives them the information they need to find the best conditions for live-cell super-resolution imaging that is sensitive to phototoxicity. This provides new fundamental principles to make live-cell SRM more stable and reliable."

Similarly, neither FRC nor RSP are new (they have been used in SQUIRREL already). It appears that this work is simply combining the two previous works of the authors – some concepts of SRRF and some of SQUIRREL – and programming them into one pipeline for searching an optimal candidate in the parameter space.

Indeed, eSRRF heavily exploits the well known and validated FRC and RSP metrics for its own optimization. We would like to highlight an important point that was less clear in the prior version of the manuscript: with the reformulation of its mathematical principles, eSRRF is able to surpass SRRF in both resolution and image quality, irrespectively of parameter optimization, thus demonstrating that eSRRF is not solely a combination of SRRF and SQUIRREL into a single pipeline. To strengthen this point, we now present a comparison between eSRRF and SRRF, where both algorithms have been optimized in a similar manner, by finding parameters that jointly optimize the FRC and RSP values. As a result, eSRRF clearly shows the capacity

to bypass SRRF in resolution and image quality (see new Suppl. Fig. 10). These features, and the new capacity for 3D, go well beyond the state of the art shown before.

None of the involved ideas are new or groundbreaking, or an important new method. At best, it is an incremental contribution and essentially just a more elaborate post-processing than before. This is also accepted by the authors in Lines 140-142 ("The increased image fidelity results from the implementation of several new and optimized routines in the radial fluctuation analysis algorithm, introduced through a full rewriting of the code."). Indeed, the approach of eSRRF is more computationally demanding as now the SRRF images need to be first constructed using each candidate in the parameter space (exhaustive search for the best set of parameters).

In practice, the parameter optimization is only performed once and can be restricted to a smaller subregion and subset of the data, only a few frames of a live-cell time-course dataset for instance. Once the optimal parameter set is defined through our optimization routine, these can be applied to any subsequent dataset without running the optimization, which represents an equivalent of time compared to the classical SRRF. Furthermore, the calculations are very fast given that the algorithm exploits state-of-the-art GPU acceleration (e.g. eSRRF processing: 1 min 26 sec for a standard 100 frames of a 512x512 pixels live-cell dataset, M=5, R=3, S=2; 5x5 Parameter sweep: 25 min 17s for 100 frames of a 512x512 pixels live-cell dataset, M=5, R=[1, 1.5, 2, 2.5, 3], S=[1,2,3,4,5]; this time can be significantly reduced to only 4 min 14 s by reducing the dataset to a representative 200x200 pixels ROI). We do not believe that the amount of time it takes to do the calculations will significantly limit researchers in applying the method. Our lab always applies a high standard for how user-friendly and applicable our tools are in practice.

Where the work is devoid generally of any new ideas over their previous works, it is strange that the authors neglect an obvious observation that their **FRC and RSP maps both appear continuous**, and an exhaustive search is not needed. I therefore rank the work quite low in any form of novelty over their previous works.

We agree that, strictly speaking, an exhaustive search is not necessary to get to the optimal reconstruction parameter set. We considered using gradient descent or similar technique here as is commonly done in optimization. But upon careful investigation, we decided against it in order to truly provide the user with a map of image quality (QnR map) which provides insights and favors a deeper understanding of the trade-offs made by any reconstruction settings. Our intention with this work is to empower the user and not brush the subtleties of the reconstruction performance under the carpet. We have decided that using an optimization algorithm that would not also empower the user (such as gradient descent) is not as impactful here. We are fully aware that it is a compromise here between algorithm execution time and richness of the output. As eluded before, we do not think the execution time is a deal-breaker here and all our beta testers were more than happy to wait for the few minutes necessary to get the QnR map to understand what the algorithm does with their data. In our view, there truly lies the value of our work.

To highlight that the QnR map evaluation requires a human-in-the-loop assessment of the reconstructions we have added Supplementary Figure 9. Here we show how the QnR maps vary locally and cases are highlighted where user intervention benefits eSRRF parameter

selection. This is particularly evident in large field-of-view images, such as the one demonstrated, which feature distinct structures and a variability in marker density. As shown in this figure, the QnR map can be nonlinear, and in the case of pronounced secondary QnR maxima, user intervention is helpful in judging which parameter set yields the best results.

To emphasize this aspect in the manuscript, we have added the following section (lines 266-275):

"While the QnR map can directly highlight optimal reconstruction settings by indicating the maximum QnR parameter combination, it also provides a window into the effect of reconstruction parameters on the output images to the user for critical analysis. Local variations in background level, emitter density, and sample structure across the field of view can cause different reconstruction requirements and non-linearities in the QnR-maps (Supplementary Fig. 9). User evaluation of QnR maps is an important component of this optimization strategy, and the parameter optimization tool is intentionally not designed to act as a black box. To aid in this evaluation, eSRRF lets users browse through the reconstructions associated with each QnR map value, enabling researchers to access the link between quality metrics and the corresponding variations in the reconstruction results."

- **3D super-resolution:** The authors claim 3D resolution improvement by applying eSRRF on fluctuations data acquired using MFM. The results in Fig. 4 show only a contrast enhancement over diffraction limited image. The dip between the structures is clearly visible in numerical sense in the diffraction limited image, and the eSRRF results only enhance the contrast by pronouncing the dip between the structures. Indeed, the result is better than the result of deconvolution (no details about how this was done are available), and this says only that the deconvolution was not applied well or was not effective. Similarly, in sup. Fig. 11(a-c), it appears again that eSRRF accomplishes only contrast enhancement.

We agree that the previous version of the manuscript didn't go into enough detail on how combining MFM and eSRRF achieves 3D super-resolution microscopy. In the new version of the manuscript, we have added realistic 3D simulations featuring varying emitter densities (Supplementary Figure 15). The simulation quantitatively shows the axial resolution achieved by 3D eSRRF and its limitations. We have also performed deconvolution for the same data, using the exact same PSF used to generate the simulation. Quantitative analysis shows that deconvolution cannot achieve the same resolution values as eSRRF. We have also added FRC and decorrelation-analysis resolution estimates for the xy and xz planes in the live-cell MFM data (Supplementary Table 4), which makes the sub-diffraction resolution more clear.

We report the deconvolution procedure as follows in the methods section:

- MFM data, line 705-707: "Deconvolution was performed with the classic maximum likelihood estimation (CMLE) algorithm in the Huygens Professional version 21.10 (Scientific Volume Imaging, The Netherlands, <http://svi.nl>)."
- Simulated data, lines 515-518: "Deconvolution of the interpolated 3D image stack was performed with the ImageJ plugin DeconvolutionLab2⁵⁷ with the PSF used for the simulation and the Richardson-Lucy algorithm (40 iterations)."

In Supplementary Figure 14 (11 in the previous version of the manuscript), we now highlight more clearly the resolution improvement achieved by eSRRF, going beyond a simple contrast enhancement. This is demonstrated through resolution estimates and intensity line profiles, quantitatively showing super-resolution both far away from the coverslip and deep inside tissue.

Significant:

- Parameters: The ranges of radius and sensitivity seems like still some parameters that the user may need to specify and need insight of. It appears that sensitivity is just some form of non-linearity. Clarification is needed on this parameter. It does not appear in the original SRRF article from 2016

The sensitivity allows for fine-tuning the PSF sharpening power applied by the RGC. In fact, the Sensitivity value S is applied as a power to the RGC map just before the temporal analysis is applied, effectively amplifying regions with high radial symmetry. This feedback helped us identify that this information was missing in the previous version of the manuscript and we have added it to Supplementary Note 2.1

- Please see line 97: "However, obtaining optimal reconstruction results ... from reconstruction artifacts and lack signal linearity." Also, line 138. These are strong statements. It will be useful to refer to some results/articles that discuss the reconstruction artifacts in fluctuation based techniques.

We have now added a reference for a comparative study pointing out artifacts created by fluctuation-based techniques (Opstad et al. 2020, arXiv, <https://doi.org/10.48550/arXiv.2008.09195>).

- Use of FRC as the resolution metric: FRC presents some challenges, as has been reported in the literature. It is highly recommended that the authors show resolution using at least one more approach. The limitation of FRC as a resolution measurement for eSRRF is evident in sup. Fig. 7. As the results in this figure show, FRC is generally an over-optimistic resolution estimate for eSRRF. It is recommended that authors back the resolution claims using **decorrelation based metric** [<https://www.nature.com/articles/s41592-019-0515-7>] in addition to FRC. For the same reason, I suggest that the resolution number reported in the graphical abstract be removed.

Indeed, FRC provides a resolution estimate that can potentially be overly optimistic, as demonstrated by Descloux et al., where the alternative "decorrelation analysis" resolution analysis was proposed. Decorrelation analysis, in turn, has also been shown to provide resolution estimates that can be artificially influenced by reconstruction parameters, as later discussed by Descloux et al., *Communications Biology*, 2021. In the context of an eSRRF parameter optimization, even if FRC resolution estimates are optimistic, they can still be used to assess relative changes in resolution and obtain the optimal reconstruction parameter set from the QnR map. This is done along with the careful controls set up to also measure the quality of reconstruction. When it comes to reporting absolute image resolution values, we now show both decorrelation analysis and FRC values for all presented figures in

Supplementary Table 4, in addition, when appropriate, we further show peak-to-peak resolution estimates by line tracing. In their majority, both FRC and Decorrelation analysis report fairly similar values, showcasing the resolution enhancement achieved by eSRRF.

- The use of RSP as a fidelity measure and nFRC as resolution metric: It is not clear what exactly is the RSP measuring that the authors refer to as fidelity. Consider in sup. Mov. 1. When $S=2$, $R=2$, then there is a lot of debris in the background. Neither do the foreground structures appear in high fidelity. But the RSP score is quite high. So, it does not seem that RSP is performing as a fidelity metric. The normalization of FRC is not clearly discussed. Does $nFRC=0$ corresponds to the lowest value of FRC in the search space and $nFRC=1$ corresponds to the highest? What if the search space is not optimally selected in the first place? Isn't it more appropriate to set $nFRC=0$ as the FRC related to the diffraction limited image?

We agree and have modified the section in the results part to improve clarity (lines 240-242):

"We use FRC to determine image resolution and Resolution-scaled Pearson correlation coefficient (RSP) as a metric for structural discrepancies between the reference and super-resolution images³¹, here referred to as image fidelity."

The RSP metric is calculated as a global correlation between the diffraction-limited reference image upsampled by interpolation and a version of the super-resolved reconstruction that was blurred to meet the same diffraction-limited resolution level. This procedure has been well described and extensively characterized in Culley et al, Nature Methods, 2018. The SSIM (structural similarity index) is an image similarity metric also extensively used to measure image reconstruction fidelity (e.g. in Zhao et al., 2020 and Chen et al., bioRxiv (2021)). RSP will highlight artifacts in the super-resolved reconstruction with a substantial weight on high amplitude differences in the actual prominent structure as they will appear in high resolution reconstructions, while low-amplitude fine background patterning has less effect on the metric. This is clearly visible in Suppl. Movie 1, where for $R=1.5$ & 2 and $S=2$ a low resolution reconstruction scores high RSP (see also Suppl. Movie 1 screen shots in Auxiliary Figure A1). For this reason, the combined QnR metric gives a more robust measure of the parameters leading to optimal image reconstructions.

Auxiliary Figure A1: Representative screenshots from Supplementary Movie 1 of the Parameter sweep output and the respective FRC, RSP and QnR maps for the eSRRF processing parameters **a** $R=1.5$ & $S=2$ and **b** $R=2$ and $S=2$.

With regard to the FRC normalization, yes, $nFRC=0$ corresponds to the lowest value of FRC in the search space and $nFRC=1$ corresponds to the highest, as the referee pointed out, and will therefore depend on the choice of the search space minimum and maximum. We understand the referee's concern here but the range can be automatically set from the FRC map itself (again, being purely data-driven as we intend the tool) and applying a mild logistic conversion to avoid discontinuity (see SI for details). It is possible to manually set the maximum FRC resolution value to the diffraction limited resolution. However, in practice, this would give more weight to lower resolution reconstructions as they are assigned higher $nFRC$ values. Indeed the results of the QnR parameter optimization depend on the effective sampling of the relevant parameter space, which should be selected in a way that the QnR maximum is not located in the periphery of the map.

Although it would be useful to relate the FRC resolutions obtained here to that of the diffraction-limited as suggested, we wanted to make the QnR map as close as possible to the resolution dynamic range of the eSRRF output so that it is as sensitive as possible to the effect of the reconstruction parameters. This is our main intention for this optimization protocol.

- **Benchmark for artefact suppression and ground truth:** In context of several figures, but more specifically sup. Fig. 4, eSRRF presents poorer contrast and presence of more details than SRRF. It is not clear if the more details in eSRRF are artifacts or whether the SRRF has suppressed these details. Data driven error analysis such as in SQUIRREL are useful. But artefacts are better studied using SEM images or other more precise and accurate imaging modalities. In this sense, the precision of STORM images is such that resolution +/- precision of them is quite close to eSRRF. In this case, using even STORM as benchmark is not useful. In the same vein for sup. Fig. 5, it does not make sense to compute the average nearest neighbor distance below the precision of STORM images. Therefore, I do not trust the numbers reported in the last 2, even the last 3 columns of this figure. In sup. Fig. 11, it is difficult to say that what is reconstructed is reliable. The question is quite important for thick samples considered in this figure, since the thick samples contribute large scattering and the details reconstructed can easily be inaccurate

The goal of optimizing eSRRF parameters is to strike a balance between image quality and resolution. To achieve this, resolution is sometimes limited to improve image quality. Conversely, reconstructions using the original SRRF method can suffer from warping and oversharpening, resulting in the illusion of high resolution in structures that do not fully represent reality (as shown in Supplementary Figure 5, formerly Figure 4). SQUIRREL error maps are particularly helpful in identifying these issues.

We agree that the best way to study artifacts is with a well-known ground truth structure, as we have previously discussed in Thevathasan et al, Nature Methods, 2019. In the previous version of the manuscript, we demonstrated eSRRF resolution enhancement in several well-known structures, including nuclear pore complexes (Supplementary Figure 8), which are widely accepted gold standards in the SMLM field. We now also show its ability to resolve calibration standards, specifically the Argo-SIM slide, as demonstrated in Supplementary Figure 3 and new simulated data (Supplementary Figure 15).

In the manuscript, we also use low emitter density SMLM data as ground truth, in which we artificially bin the temporal data to increase emitter density and demonstrate how eSRRF reconstructions are capable of recovering high-density resolution. In the previous version of the manuscript, we mistakenly referred to this data as STORM, but it is actually DNA-PAINT data. We have corrected this error. While we agree with the reviewer that the precision of STORM images may have a resolution/precision similar to that of eSRRF, this analysis serves a different purpose. With this data, we demonstrate that eSRRF can generate super-resolution data similar to low-density STORM, but with considerably fewer frames and higher blinking densities. This demonstration is crucial to establish that the algorithm can handle high-density data accurately, as is often required in fast (live-cell) imaging, where the number of available time frames for reconstruction is limited. To facilitate this evaluation, we report the resulting emitter density in the binned raw data as the average nearest-neighbor distance in Supplementary Figure 6 (formerly Figure 5). Furthermore, we believe that we now present a large set of complementary data in reference structures that demonstrate the degree of fidelity in similar reconstructions.

We also agree that Supplementary Figure 14 (formerly Figure 11) lacked quantitative metrics of the reconstruction performance. All eSRRF reconstructions were performed using QnR parameter optimization to determine the best reconstruction parameters, minimizing reconstruction artifacts caused by scattering in large samples, such as the zebrafish embryo. However, this also limits the level of detail visible deep inside the sample. In the updated version of the figure, we have included resolution metrics and intensity profiles. With this additional quantification, we demonstrate the super-resolution capabilities of eSRRF even deep inside tissue, far away from the coverslip.

- Rolling window: In sup. Fig. 10, two consecutive time windows have 90 frames in common. How do we pin the time of ER formation in this case? How do we know that it is about formation and not coming into the imaging plane from out-of-focus? Also, there are quite a few discrepancies in this figure. In the caption, the temporal sampling is reported as 10 Hz, whereas in the main text, it is reported as 50 ms (i.e. 20 Hz). Image section width is reported as 3 μm , whereas the scale bar is shown as

2 μm (which is visibly more than 2/3rd of the image section). I suggest that the authors check these details.

Thank you for pointing out the mistake in the figure and the corresponding parts of the manuscript. Indeed, the temporal sampling is 10 Hz, and the supplementary figure was accidentally cropped, creating the illusion of a mismatch between image width and scale bar size. We have corrected this in the new version of the manuscript.

Also, the title of the corresponding supplementary figure (which used to be Supplementary Figure 10, but is now Supplementary Figure 13) was changed to reflect the fact that there is no way to tell ER formation or rearrangement apart from planes that are out of focus. Moreover, it's a limitation of the imaging system (2D imaging) and not of the algorithm used here to super-resolve the lateral spatial content.

The concept of rolling window analysis or interleaved reconstruction has been previously demonstrated for SIM (Ma et al., 2018, doi: 10.1002/jbio.201700090 & Guo et al., 2018, doi: 10.1016/j.cell.2018.09.057). With this approach, the effective frame rate is increased, which can lead to an increase in temporal resolution. However, the increase in temporal resolution does not occur at the same rate, and highly dynamic structures are still subject to movement artifacts. Nevertheless, as shown by Ma et al., the approach still exceeds the performance of temporal interpolation. As the information for each reconstruction still originates from a frame window that exceeds the rate of temporal sampling, it is not possible to assign an exact timepoint to an event observed in a single reconstructed frame. However, the velocity of dynamic changes within a frame sequence can still be assessed with interleaved reconstruction. In the kymographs presented in Auxiliary Figure A2 below this is nicely demonstrated. While the linear analysis only provides three observation timepoints (no extrusion > half way > fully connected) with the rolling window it is possible to see how this new connection is formed in detail.

Auxiliary Figure A2: Temporally color coded representation of dynamic rearrangement of the ER in COS-7 cells expressing PrSS-mEmerald-KDEL imaged in HiLO and processed with eSRRF (left) and kymograph representations of the region marked by the white rectangle of a time series processed linear temporal analysis with a window of 100 frames (middle) and with a rolling window analysis with intervals of 10 frames (right). See also Supplementary Figure 13.

Minor:

20

- Line 168: how was the temporal binning performed. Was it binning consecutive frames together or in an interleaved fashion? In the first case, photobleaching introduces sparsity in the later frames. In the second case, the signal to noise ratio is relatively uniform across the binned frames.

Auxiliary Figure A3: The number of localizations per frame over the course of a 50000 frame DNA-PAINT acquisition remains constant throughout the experiment with an average of 144 localizations per frame.

The binning was performed by summing consecutive frames. The SMLM ground truth data is based on a DNA-PAINT experiment where there is negligible photobleaching or decline in detections, as shown in the plot above (Auxiliary Figure A3). In the new version of the manuscript, we have added more information about the localization density and the strategy for grouping (lines 543-546):

"To create datasets with increased emitter density, temporal binning was performed by summing sub-stacks of the SMLM image sequence. The raw data has an average emitter density of 0.121 localisations per frame and μm^2 which remains constant throughout the whole DNA-PAINT acquisition."

- The authors just coin the term QnR, not providing any elaboration of it. Is it an acronym? What does it stand for?

Thank you for picking this up; we corrected this error in the revised version of the manuscript as the acronym was not introduced before. QnR stands for "Quality and Resolution" and is a metric that represents the compromise between image reconstruction quality and resolution as measured by FRC and RSP scores.

- In Fig. 3, the use of NSTORM in (a) and Fast-GI TIRF in (b) is quite confusing. Either it is incorrect annotation, or this annotation was done for some other purpose, or the images need to be checked.

Indeed, these labels are not relevant, and we removed them from the figure.

- In sup. Tab. 3, it will be useful to include optical parameters of the systems, diffraction limited resolution, SRRF resolution, and eSRRF resolution in each case.

We've added information about the imaging and processing parameters as well as estimates of the images' resolution to the supplementary table, which is now called Supplementary Table 4.

- In the simulated examples, more details are needed. In sup. Fig. 3, what is the distance between emitters (seems larger than the claimed resolution) in the third circle (where emitters are not resolved by eSRRF). What is the distance between the circles?

We have added the information to the respective figure caption and method description (lines 477-478):

"Simulated ground truth indicating the positions of individual molecules placed on concentric rings with radii increasing by 220 nm steps. On each ring the molecules are separated by 57.5, 115, 173, 230, 288 and 345 nm, respectively."

- In sup. Fig. 11b, Collagen and F-actin seems color coded, but inclusion of them as different texts in the two planes seems misleading. The upper plan has text F-Actin written on it and the lower plan has Collagen. However, it seems both the planes show both collagen and F-actin.

Thank you for pointing this out. We have corrected labels in the panel b of supplementary figure 14 to avoid this misconception. All images in panel b display F-Actin and Collagen I.

- In sup. Fig. 12, it would have been useful to indicate the plane in which the different lines lie, so that appreciation of the sub-diffraction distance between the different lines is easier.

In the new version of the manuscript we have exchanged supplementary Figure 15 for a more complex evaluation of the 3D super-resolution power of eSRRF. Here, we have also added a detailed description and a schematic representation of the 3D model architecture.

Reviewer 3:

I read the eSRRF manuscript (including the corresponding bioRxiv preprint version) by Professor Henriques and co-workers with great interest. The main novel developments include: 1) Updated SRRF algorithm plus adaptive parameters (radius, sensitivity, number of frames) choosing for high-fidelity or balanced fidelity-resolution reconstruction; 2) 3D SRRF realization by extending the previous SRRF theory to axial dimension and incorporating multi-focus imaging. However, regretfully, although the manuscript does have some novelty and contribution, I am afraid it is not sufficient and thus couldn't recommend its publication on high-quality journals including Nature Methods.

- 1) After it was first proposed by Henriques' group, SRRF method, including the corresponding algorithm, has been followed and improved by several groups. However, the current manuscript neither includes any literature review in the Introduction, nor performs any simulation or experimental comparison between the proposed eSRRF algorithm (under the situation without adaptable parameter searching) and the other developments.

To our knowledge, there has been no improvement to the SRRF concept itself. However, there are several implementations where the SRRF method is combined with new sample preparation techniques or with other super-resolution imaging approaches (joint tagging approach with QDs (Zeng *et al.* 2018), Expansion Microscopy (ExM) - (Shaib *et al.* 2022) and ExM with Airyscan imaging (Want *et al.* 2020)). There are also approaches that introduce additional data treatment steps to achieve artifact reduction, such as cross-cumulant analysis (Zeng *et al.* 2020) and gradient variance analysis (Gong *et al.* 2021). All of these concepts are now listed and compared in the newly added Supplementary Table 2 and can benefit from being directly combined with eSRRF as well.

Although all these variations of our original approach are valid and highlight the enthusiasm around the method, we do not feel that these represent the same breakthrough that we propose here, especially from the point of view of tackling the main limitations of any fluctuation based methods, namely that artifacts can be introduced. Our data-driven and human in the loop approach here truly represent a shift in how we can, as a community, better understand the potentials of fluctuation-based super-resolution.

In the revised version of the manuscript, we added the following section to the introduction to help readers find other reference studies that build on or use SRRF (lines 98-103):

"Since its inception, several adaptations of SRRF have been proposed by the community based, for example, on the combination with other advanced imaging approaches^{26,27} or on the introduction of additional data preprocessing steps²⁸ (see Supplementary Table 2 for a summary), highlighting the interest in and potential impact of the method on the imaging community."

- 2) Did the authors implement the parameter exploration scheme into their original SRRF reconstruction? I would like to see the comparison among SRRFm reconstruction with and without QnR parameter searching, and also eSRRF reconstruction with and

without QnR parameter searching, which will help distinguish which steps improve the resolution or fidelity.

Thank you for the excellent suggestion. We have added a new Supplementary Figure 10 where we implement the parameter optimization for SRRF. A comparison between a parameter optimized SRRF and eSRRF shows that even in this scenario, eSRRF is able to achieve higher resolution and quality values. This is possible due to the new mathematical improvements integrated into the eSRRF analytical engine.

- 3) How to evaluate the parameters based on automated QnR searching is “optimal” of not? This is different from the optimal PSF design in 3D SMLM based on CRLB metric.

The QnR metric is very different from the CRLB, as it is just weighting two metrics. Indeed, in certain cases, the user might want to attribute more weight to resolution than to fidelity. In this scenario, QnR values, although informative, can always benefit from a user critical analysis. QnR is a metric to estimate the best joint reconstruction resolution and quality that can be experimentally achieved, unlike the CRLB, which allows one to calculate the theoretical maximum.

- 4) I agree that high-fidelity imaging is more and more urgent with the super-fast development of super-resolution imaging field. However, the strategy of automated reconstruction parameter search and finetune based on some metrics to improve image fidelity has been already used in many fields, including the super-resolution imaging. Typically, neural network training with the feedback of loss functions.

To our best knowledge, the concept of QnR maps, as a tool for eSRRF parameter optimization, is completely novel. Even though the measures of quality (RSP) and resolution (FRC) are gold standards in the field, they have never been iteratively combined to find the best balance between image fidelity and resolution. Other works have only focused on assessing the two aspects separately, for example: Schäfer et al., *Journal of Microscopy* (2004), Ball et al., *Scientific Reports* (2015), Zhao et al., *bioRxiv* (2022), <https://doi.org/10.1101/2022.12.12.520072>). Subjective, user-defined image quality scores have also been used in other studies (Durand et al., *Nat Comm* (2018)). To our knowledge, there is only one approach that considers both image quality and resolution by employing a multicomponent loss function to train an ESRGAN for deep learning based super-resolution (DL-SRM) image reconstruction (Chen et al., *bioRxiv* (2021), <https://doi.org/10.1101/2021.10.08.463746>). In contrast to DL-SRM, the data-driven eSRRF parameter optimization does not depend on a pretrained model, prior knowledge on the dataset, simulated ground truth, or annotated data, basing its scoring on the direct unsupervised analysis of the data being collected via FRC and RSP calculations. This procedure is both simpler and more robust when the data being collected has new features or properties not observed before. Furthermore, the implemented parameter optimization based on the QnR metric directly provides insight into quantitative and comprehensible measures of image fidelity and resolution, enabling the user to critically analyze the results and take an informed decision.

- 5) The 3D SRRF has been already achieved by another group [1,2].

We thank the reviewer for the additional information. We have updated the manuscript to include the references and emphasize the differences between eSRRF and other approaches for 3D super-resolution imaging.

The mentioned publications present powerful tools for volumetric live-cell super-resolution imaging. Both approaches are based on light sheet illumination and sequential acquisition of the axial planes and, indeed, they are based on extending the original SRRF algorithm into 3D. In these articles the adapted SRRF algorithm was extended to calculate 3D radial symmetry, the reconstructions achieved display a high level of artifacts and artificial sharpening, as acknowledged by the authors. With radial fluctuation Bayesian analysis (RFBA) Chen et al. further demonstrated the capacity to significantly accelerate 3B analysis in 3D by using a rough super-resolution estimate generated by the 3D SRRF as a starting point. Only when combined with 3B can high quality super-resolution be achieved. The use of 3D eSRRF might be able to speed up and improve RFBA even more. By adding deep learning to selective plane illumination microscopy, in Zhao et al. authors were able to achieve fast volumetric super-resolution imaging. While they use 3D SRRF to create ground truth super-resolution data of fixed samples to train the network, the fast volumetric live-cell super-resolution imaging is based on deep learning based SRM. Reference to these works can now be found in Supplementary Table 5. There, we compare and contrast alternative volumetric live-cell super-resolution imaging approaches.

- 6) The authors demonstrate the estimation of the number of frames necessary for SRRF reconstruction based on tSSIM metric. However, this can only be achieved in livecell experiments, how to determine the frame parameter in general fix-cell imaging?

There are two different aspects to consider: 1) In the case of live-cell imaging, the tSSIM metric allows for the estimation of the maximum number of frames in a frame window that still avoids significant movement artifacts; 2) In the case of fixed samples, varying the frame window size for the temporal analysis in the eSRRF parameter sweep allows researchers to estimate the minimum or sufficient number of frames for optimal sampling, ensuring a good quality reconstruction result.

We have extended the respective section in the main text to make this more clear (line 281-298):

"An important aspect of live-cell super-resolution imaging is its capacity for observation and quantification of dynamic processes at the molecular level. With respect to fluctuation based methods, there are two aspects to be considered when it comes to observing fast dynamics. As each super-resolved reconstruction is based on processing a stack of hundreds or more images any dynamic changes happening within this frame window will lead to motion blur. On the other hand, reducing the number of frames for the super-resolved reconstruction can compromise the reconstruction quality. To address this aspect and provide an estimate of the optimal frame window, we have integrated temporal structural similarity (tSSIM) analysis into the eSRRF framework. Here, we calculate the progression of the structural similarity⁵³ at the different time points of the image stack relative to the first frame (Supplementary Fig. 11). This allows us to identify the local molecular dynamics (Supplementary Fig. 12) and estimate the maximum number of frames within which the structural similarity is retained, meaning that

there is no observable movement (Fig. 2d). By combining tSSIM with eSRRF, we can determine an optimal number of frames required for the reconstruction to recover dynamics, while reducing motion blur artifacts (Fig. 2e). The tSSIM is complemented by the parameter optimization tool, which jointly aids in finding the optimal Radius and Sensitivity parameter sets, and allows testing of different frame window sizes. This enables the user to identify the minimum number of frames to be analyzed or even acquired to ensure a good quality reconstruction”

- 7) The section “eSRRF works across a wide range of live-cell imaging modalities” doesn’t provide any new information as the universal characteristic of SRRF reconstruction applied to different modalities has already been shown, including their first Nature Communications SRRF paper.

In our view, it is still important to show that this universal applicability is still valid for eSRRF and spans a variety of different imaging modalities. These range from fast 2D (TIRF and HILO imaging) to volumetric imaging in a lattice light-sheet microscope. Towards this goal, we also show that eSRRF works with a number of fluorescence labeling methods that are live cell friendly. These include fluorescent protein (FP) tagging with eGFP/mEmerald, optimized photoswitching FPs (ffDronpa and SkylasS), organelle-specific live-cell dye staining (BODIPY ER-Tracker), and transient expression of self-labeling protein tags (Halo Tag) conjugated proteins targeted by cell permeable organic dyes (JF549).

- 8) The authors claimed “live-cell SDC and LLS imaging” in the text, however, there is no time-lapse data shown in the figure or video (only time-lapse TIRF-eSRRF data). In my understanding, SRRF will hugely increase the acquisition time in 3D scanning as hundreds to thousands of frames need to be acquired in each axial plane.

Extending LLS imaging for eSRRF will require the acquisition of ~100 frames per axial plane to reconstruct a single super-resolved volume, significantly slowing down the effective imaging speed. While in this case the imaging speed might not be sufficient to ensure the temporal resolution needed to observe fast dynamics, eSRRF can still provide super-resolution imaging in living cells and map intracellular structure while bypassing the typical distortions associated with fixation artifacts. We have revised the wording in the new version of the manuscript to be more conscious and precise about data where we just demonstrate super-resolution imaging snapshots of living cells, in tandem to highlighting figures and videos in which we actually present live-cell super-resolution time lapse data (Suppl. Movie 1, Figure 3b & Suppl. Movie 4, Figure 4 & Suppl. Movie 5)

- 9) The explanation about the implementation of MFM into SRRF is too little to be understood well. How is this achieved experimentally? What is the hardware system like? This is one of the main contribution of the manuscript and thus more details should be shown.

In the new version of the manuscript we have added Supplementary Note 3 with an extensive description of the MFM setup and experimental procedures as well as 3D registration and processing.

- 10) The reconstructed depth of MFM-SRRF is relatively shallow, 3.6 μm . What's the limit?

The depth of the MFM acquisition is limited by the number of focal planes (here, 9 planes) and the maximal plane distance (here, 400 nm). These parameters are selected to ensure optimal axial sampling of the PSF. To add more focal planes, one would need a multifocus grating that was purposely redesigned for that purpose and a larger lens array. A limiting consideration is the additional reduction of signal per plane, which degrades reconstruction quality. Other multifocus systems are generally limited to comparable axial ranges, for example: in 3D SOFI (65x65x3.5 μm^3 using prisms yielding 8 z-planes, Geissbuehler et al. (2014)) or SOLIS (2 μm thick instant-volumes, Ströhl et al. 2022).

- 11) 3D live mitochondria experiments. The comparison between the eSRRF (Figure 4) and deconvolution (Supplementary Fig. 14) results confuse me a lot. The selected xy (i) and xz (ii) intensity profiles are too subjective and unconvincing (including other figures, such as Figure 3, Supplementary Fig.2.). In my view, eSRRF and deconv results are very similar and can't provide me with obvious difference. Actually, I even think deconv result performs better.

Deconvolved images may appear appealing due to their smoothly represented structures, but they do not outperform eSRRF in terms of image resolution. In the previous version of the manuscript, this was already shown with representative line profiles and resolution metrics. In the new version of the manuscript, we now also use decorrelation analysis to comparatively measure the resolution of the eSRRF reconstructions against deconvolution. We have now added a set of simulated 3D data that was analyzed by both eSRRF and deconvolution (Supplementary Fig. 16) and compared. Even with simulated data, where we use the same point spread function (PSF) to create and deconvolve the image stack, eSRRF achieves a better resolution.

- 12) Movie S4. Within the displayed short imaging duration of 17s, it seems that the photobleaching and phototoxicity are pretty severe as the whole mitochondria network contracts quickly. The mitochondria also become fewer and fewer and more fragmented. This concerns me about the practicality of this method.

In the new version, we have added two new datasets to demonstrate live-cell friendly imaging compatibility. One observes actin rearrangement at super-resolution level over 12 hours in U2OS cells (Supplementary Movie 4). The second data set is a 3D super-resolved view of mitochondria network fluctuations in U2OS cells acquired by MFM over the course of >3 min. To ensure cell health during this long volumetric acquisition, we reduced the excitation intensity, sacrificing resolution to some degree (FRC values are now estimated to be ~125 nm), but still achieving a resolution improvement of 2.5-fold over diffraction (Supplementary Movie 5).

- 13) The temporal resolution and duration of (3D) (e)SRRF shown in the manuscript is too low considering the rapid development of the ultrafast and long-term superresolution imaging field, either in 2D or 3D, such as Hessian SIM, GI SIM, AiryScan, IDDR-SPIM, 3D pRESOLFT, etc., although one may argue that SRRF has some advantages.

Indeed, the manuscript would benefit from a comparison between (3D) eSRRF and other comparable methods, in the context of live-cell super-resolution microscopy. An important point to note is that we do not claim unsurpassed temporal or spatial resolution with eSRRF, there are techniques, like Hessian SIM, GI SIM, which are able to achieve faster super-resolution imaging, but also require specialized optical hardware, unlike eSRRF for the 2D case. 2D eSRRF allows fast super-resolution imaging on microscopy platforms that are widely accessible to users without posing any sophisticated hardware requirements. We therefore limit the comparison to other fluorescence fluctuation analysis based approaches (Supplementary table 1). When it comes to 3D super-resolution imaging in living cells, the temporal resolution of ~1 Hz/volume achieved with 3D eSRRF is only surpassed by IDDR-SPIIM which is a deep learning based 3D super-resolution approach on specialized hardware (Supplementary table 5). As already discussed in point 4, deep learning based SRM is very powerful but also has its drawbacks.

- 14) The reconstructed results confused me often. In some cases, it seems that SRRF images have better resolution in spite of the fidelity (following figure (a, c, d)), while in other cases, eSRRF outperforms (following figure (b)).

As the goal of eSRRF parameter optimization is to find the best balance between image quality and resolution, it will sometimes limit the resolution improvement to keep the reconstruction as good as possible. The original SRRF reconstruction, on the other hand, contains warping and oversharpening artifacts that could be misinterpreted as high resolution structures (as shown in Supplementary Figure 5/formerly 4). These artifacts can be well identified by using well known ground truth structures and robust image quality metrics. Supplementary Figure 3 now shows a comparison between SRRF and eSRRF using the Argo-SIM calibration slide.

This data both shows that SRRF is sensitive to producing warping artifacts, which are not present in eSRRF, and that eSRRF is able to surpass the resolution of SRRF while maintaining higher image quality.

Minor comments.

1) RSP metric should be explained at its first appearance.

We have added the following section to the manuscript: "...Resolution-scaled Pearson correlation coefficient (RSP) as a metric for structural discrepancies between the reference and super-resolution images, also called image fidelity."

2) Scale bar is missed in Fig.1

We have corrected this mistake.

3) Fig. 1b. Conventional SRRF reconstruction steps should be added to the figure for better comparison.

We now more extensively compare SRRF and eSRRF, including parameter optimization for both algorithms (Supplementary Figure 10).

4) Fig. 1c. FRC resolution of SMLM is 71nm. Why the DNA-PAINT resolution is so low, which doesn't matches my experimental experience.

We measure the same resolution level with FRC and Decorrelation analysis. Indeed, it might be possible to achieve higher-resolution using DNA-PAINT, however, we believe the data we present is a good representation of common experiments by researchers.

5) Fig. 2d-e are missed (preprint version has).

We are sorry about this mistake. The Figure was accidentally cropped, we have corrected this in the new version of the manuscript.

6) Supplementary Movie 4. The note should be "eSRRF" rather than "SRRF". Besides, the same period is repeated several times.

We have corrected the mistake.

[1] Chen, Rong, et al. "Efficient super-resolution volumetric imaging by radial fluctuation Bayesian analysis light-sheet microscopy." *Journal of biophotonics* 13.8 (2020): e201960242.

[2] Zhao, Yuxuan, et al. "Isotropic super-resolution light-sheet microscopy of dynamic intracellular structures at subsecond timescales." *Nature Methods* 19.3 (2022): 359-369.

The authors thank both the editorial team and reviewers for their consideration and feedback.

Decision Letter, first revision:

Dear Ricardo,

Thank you for submitting your revised manuscript "High-fidelity 3D live-cell nanoscopy through data-driven enhanced super-resolution radial fluctuation" (N METH-A48988B). Sorry for the delay on our end. We were waiting on one reviewer. It has now been seen by an original referee and their comments are below. The reviewers find that the paper has improved in revision, and therefore we'll be happy in principle to publish it in Nature Methods, pending minor revisions to comply with our editorial and formatting guidelines.

We are now performing detailed checks on your paper and will send you a checklist detailing our editorial and formatting requirements in about two weeks. Please do not upload the final materials and make any revisions until you receive this additional information from us.

TRANSPARENT PEER REVIEW

Nature Methods offers a transparent peer review option for new original research manuscripts submitted from 17th February 2021. We encourage increased transparency in peer review by publishing the reviewer comments, author rebuttal letters and editorial decision letters if the authors agree. Such peer review material is made available as a supplementary peer review file. Please state in the cover letter 'I wish to participate in transparent peer review' if you want to opt in, or 'I do not wish to participate in transparent peer review' if you don't. Failure to state your preference will result in delays in accepting your manuscript for publication.

ORCID

Sincerely,
Rita

Rita Strack, Ph.D.
Senior Editor
Nature Methods

Reviewer #3 (Remarks to the Author):

The reviewer thanks authors for the huge efforts of revising the original manuscript, providing new data, and addressing the comments raised by our reviewers. I think that the major comments from myself and the other reviewer are almost the same, that is, the novelty, reconstructed quality of their new algorithm, and the advantages and advances over the existing techniques. The authors now restrict the novelty more to the point of the balance between image fidelity and resolution (which means image resolution may be lower but with higher fidelity in some scenario), and provided detailed comparison of the pros and cons of these techniques. I agree that this might be an important reminder to the super-resolution community that resolution is not always the top priority and our sole pursuit, especially when the resolution of optical microscopy has been pushed to 1 nm region, and considering from a user not technique developer point. Considering this, I can recommend its publication and look forward to seeing the synchronous developments of other imaging parameters, such as speed, depth, background, quantitatively, etc. I wish optical microscopy, especially super-resolution microscopy, could be more and more powerful by developing from not only single resolution but also diverse aspects and benefit more users.

Author Rebuttal, first revision:

Dear Reviewers and Editorial Team at Nature Methods,

We are grateful for your time and effort in reviewing the new version of the manuscript entitled "High-fidelity 3D live-cell nanoscopy through data-driven enhanced super-resolution radial fluctuation". We are very happy to know that we were able to address most of the concerns raised

and we were able to significantly improve our work based on your insightful feedback and constructive criticism. To address the remaining reviewer comments we have compiled a point-by-point response below.

Our response to the Review in blue:

Reviewer #1:

Remarks to the Author:

I acknowledge that the authors have put considerable effort into revising the manuscript and providing additional results. However, my original technical concerns remain largely unaddressed and I painstakingly put forth my comments again in the light of the revision done by the authors.

It appears that the authors even strongly claim ‘new fundamental principles in SRM’ and ‘redefining the fundamental principles of SRRF’, without actually substantiating them. I elaborate the two different points below.

While claiming ‘new fundamental principles’ in SRM, they essentially refer to defining QnR which is a function of two quality indicating parameters (nFRC and RPC here) in the form $2ab/(a+b)$. Such joint score for two metrics is actually very common, for example the F-score that merges recall and precision of a classification model into a single metric. I do not see any new fundamental principle here. If the authors think that proposing QnR is the new fundamental principle being put forth, then instead of defending it on SRRF’s modification, I would have liked to see that the authors show that this principle is generally applicable to a huge variety of SRM approaches, including other fluctuations based techniques, single molecule reconstruction algorithms, and structured illumination microscopy algorithms. If the authors want to focus only on eSRRF, then I recommend that they do not claim putting forth a ‘new fundamental principle’ in SRM.

The reviewer raises a fair criticism that the use of a joint metric like QnR is not fundamentally new. We agree that similar approaches combining multiple metrics exist in other fields. However, the specific application of balancing resolution and fidelity in a single score for super-resolution image reconstruction quality is, to our knowledge, novel. While the QnR concept could be extended to other super-resolution modalities, our focus here has been on presenting and validating it in the context of eSRRF.

While claiming ‘redefining the fundamental principles of SRRF’, it is not clear to me which fundamental principles of SRRF were redefined. I suppose supplementary note S2 (separated into S2.1 and S2.2) was supposed to explain which specific principle were redefined, but despite multiple readings of this note I am unable to point out what is fundamentally redefined. In this note, the authors claim that eSRRF ‘uses the knowledge of PSF and the imaging system’, which

I do not see either. I see that the authors define σ_0 based on R , which is defined by the authors in this note as the FWHM of the PSF of the system. But R is a sweepable parameter in eSRRF, and therefore the R that indicates the highest QnR or the R corresponding to the choice made by the user may not correspond to the actual PSF of the system. Also, it is impossible to see how the imaging system (for example TIRF or HiLo or whatever) was explicitly used in eSRRF. Instead, my understanding is that the authors have designed eSRRF to actually become deliberately blind to any information of the imaging system or its PSF at least for all the example systems, except MFM.

In Supplementary Note S2, we provide a detailed explanation of the principles behind the eSRRF reconstruction technique. However, for the sake of readability and clarity, we have refrained from pointing out the differences between eSRRF and SRRF. Instead, in the first section of the results part of the main manuscript titled "eSRRF provides high-fidelity SRM images", we highlight the differences between the two techniques. Besides a more robust interpolation method, eSRRF introduces a weighting map that allows for more efficient exploitation of the local environment of the pixel of interest based on the imaging system's Point Spread Function (PSF).

On the other big concern that I have, the authors seem to be confused between whether eSRRF reduces/minimizes the user bias or whether it guides the user to make informed decisions. That the user anyway has a decision to make by clicking several boxes in the QnR map (and/or the nFRC and RPC maps) clearly implies that user will make a final selection based on personal bias. It may be biased towards qualitative appearance, or towards a number that represents better resolution, or a number that represents better RPC, or a number that represents better QnR. In fact, I think it puts the pressure on the user to learn many more technical terms and their implications in the quality of image, thereby introducing more biases, rather than simplifying things for them. Also, nFRC, RPC, QnR may indicate some quality metrics (mostly aesthetic in my opinion) but have no consequence or correlation to how reliable the final image is for inference.

You make a fair criticism that the eSRRF optimization still requires some user decision-making and, therefore does not completely eliminate bias. Our intention is to provide quantitative metrics to guide the user towards an informed decision rather than removing human judgment entirely. While nFRC, RPC, and QnR may not guarantee reliability for downstream analysis, they do aid in selecting parameters that balance resolution and artifacts. We agree that more terminology is introduced, which could be burdensome for new users. To mitigate this, we have expanded the user guides to clearly explain each metric's meaning and tradeoffs (<https://github.com/HenriquesLab/NanoJ-eSRRF/wiki/Parameter-sweep>):

"The FRC is a quantitative metric that allows to determine the image resolution of an eSRRF reconstruction. The Resolution-scaled Pearson correlation coefficient (RSP) serves as a metric

for structural discrepancies between the reference and super-resolution images, which is also referred to as image fidelity.

The FRC and RSP sweep map output can be used to calculate the Quality and Resolution (QnR) metric map, combining the two image quality metrics to find a compromise between image fidelity and resolution.

The intention here is to provide quantitative metrics to guide the user towards an informed decision while keeping the human-in-the-loop. It is very important to consider as a user that, while the QnR map can directly highlight optimal reconstruction settings by indicating the maximum QnR parameter, local variations in background level, emitter density, and sample structure across the field of view can cause different reconstruction requirements and non-linearities in the QnR-maps.

Therefore, a critical analysis of the reconstruction results of the sweep range by the user is mandatory.

By guiding users through quantitative optimization, eSRRF aims to reduce bias relative to manual parameter tuning. Keep in mind that FRC, RPC, and QnR are supporting you in selecting parameters that balance resolution and artifacts, however depending on the data and question at hand you might want to prioritise specific subregions of the image. We recommend saving the sweep results and reporting the sweep range and the chosen reconstruction parameters.”

Ultimately, some human bias is unavoidable in assessing image quality. However, by guiding users through quantitative optimization, eSRRF aims to reduce bias relative to manual parameter tuning. Thank you for raising this important point - we have clarified the limitations of fully automating optimization for the users.

PS> At several places, the supplementary figures are being referenced incorrectly. For example, In Note S1, the supplementary figures to be referred are Fig. S4, not S3. Please correct for these details.

Thank you for spotting these mistakes. We have corrected the errors.

Reviewer #3:

Remarks to the Author:

The reviewer thanks authors for the huge efforts of revising the original manuscript, providing new data, and addressing the comments raised by our reviewers. I think that the major comments from myself and the other reviewer are almost the same, that is, the novelty, reconstructed quality of

their new algorithm, and the advantages and advances over the existing techniques. The authors now restrict the novelty more to the point of the balance between image fidelity and resolution (which means image resolution may be lower but with higher fidelity in some scenario), and provided detailed comparison of the pros and cons of these techniques. I agree that this might be an important reminder to the super-resolution community that resolution is not always the top priority and our sole pursuit, especially when the resolution of optical microscopy has been pushed to 1 nm region, and considering from a user not technique developer point. Considering this, I can recommend its publication and look forward to seeing the synchronous developments of other imaging parameters, such as speed, depth, background, quantitatively, etc. I wish optical microscopy, especially super-resolution microscopy, could be more and more powerful by developing from not only single resolution but also diverse aspects and benefit more users.

Thank you for the very positive response and for highlighting this important point.

Final Decision Letter:

Dear Ricardo,

I am pleased to inform you that your Article, "High-fidelity 3D live-cell nanoscopy through data-driven enhanced super-resolution radial fluctuation", has now been accepted for publication in Nature Methods. Your paper is tentatively scheduled for publication in our December print issue, and will be published online prior to that. The received and accepted dates will be April 21, 2022 and September 29, 2023. This note is intended to let you know what to expect from us over the next month or so, and to let you know where to address any further questions.

Over the next few weeks, your paper will be copyedited to ensure that it conforms to Nature Methods style. Once your paper is typeset, you will receive an email with a link to choose the appropriate publishing options for your paper and our Author Services team will be in touch regarding any additional information that may be required.

You will receive a link to your electronic proof via email with a request to make any corrections within 48 hours. If, when you receive your proof, you cannot meet this deadline, please inform us at rjsproduction@springernature.com immediately.

Please note that *Nature Methods* is a Transformative Journal (TJ). Authors may publish their research with us through the traditional subscription access route or make their paper immediately open access through payment of an article-processing charge (APC). Authors will not be required to make a final decision about access to their article until it has been accepted. [Find out more about Transformative Journals](https://www.springernature.com/gp/open-research/transformative-journals)

Your paper will now be copyedited to ensure that it conforms to Nature Methods style. Once proofs are generated, they will be sent to you electronically and you will be asked to send a corrected version within 24 hours. It is extremely important that you let us know now whether you will be difficult to contact over the next month. If this is the case, we ask that you send us the contact information (email, phone and fax) of someone who will be able to check the proofs and deal with any last-minute problems.

If, when you receive your proof, you cannot meet the deadline, please inform us at rjsproduction@springernature.com immediately.

Once your manuscript is typeset and you have completed the appropriate grant of rights, you will receive a link to your electronic proof via email with a request to make any corrections within 48 hours. If, when you receive your proof, you cannot meet this deadline, please inform us at rjsproduction@springernature.com immediately.

Once your paper has been scheduled for online publication, the Nature press office will be in touch to confirm the details.

Once your paper has been scheduled for online publication, the Nature press office will be in touch to confirm the details.

Content is published online weekly on Mondays and Thursdays, and the embargo is set at 16:00 London time (GMT)/11:00 am US Eastern time (EST) on the day of publication. If you need to know the exact publication date or when the news embargo will be lifted, please contact our press office after you have submitted your proof corrections. Now is the time to inform your Public Relations or Press Office about your paper, as they might be interested in promoting its publication. This will allow them time to prepare an accurate and satisfactory press release. Include your manuscript tracking number NMETH-A48988C and the name of the journal, which they will need when they contact our office.

About one week before your paper is published online, we shall be distributing a press release to news organizations worldwide, which may include details of your work. We are happy for your institution or funding agency to prepare its own press release, but it must mention the embargo date and Nature Methods. Our Press Office will contact you closer to the time of publication, but if you or your Press Office have any inquiries in the meantime, please contact press@nature.com.

Nature Portfolio journals [encourage authors to share their step-by-step experimental protocols](https://www.nature.com/nature-research/editorial-policies/reporting-standards#protocols) on a protocol sharing platform of their choice. Nature Portfolio 's Protocol

Exchange is a free-to-use and open resource for protocols; protocols deposited in Protocol Exchange are citable and can be linked from the published article. More details can be found at www.nature.com/protocolexchange/about.

Best regards,
Rita

Rita Strack, Ph.D.
Senior Editor
Nature Methods